EMBO
Molecular Medicine

# Vascular malformations: from genetics to therapeutics

Gabriel Morin [1,2,3,4,5], Ilaria Galasso[1,2,5] & Guillaume Canaud [1,2,3 ✉]

## Abstract

**Vascular malformations (VMs) are congenital disorders characterized by structurally abnormal blood and lymphatic vessels. Advances in genetics have revealed that most sporadic VMs result from post-zygotic variants in genes involved in key endothelial signaling pathways, including the phosphoinositide-3-kinase (PI3K) and the mitogen-associated proliferation kinase (MAPK) pathways. As these variants are shared with cancer, genetics now have theragnostic impact by helping predict relevant targeted therapies. mTOR and PI3Kα inhibitors such as sirolimus and alpelisib have shown promising efficacy in slow-flow VMs, while reports have suggested that MAPK inhibitors such as trametinib may improve arteriovenous malformations. Despite these advances, several challenges remain, including obtaining accurate genetic diagnosis, enhancing treatment efficacy while mitigating drug-related toxicities, and personalizing multimodal treatment strategies. Emerging approaches such as mutant-selective inhibitors, proteolysis-targeting chimeras, and gene therapy hold promises for improving treatment specificity and minimizing adverse effects. This review provides an overview of the genetic bases of VMs, recent advances in targeted therapies, and future directions in the field, highlighting the ongoing evolution of precision medicine for VMs.**

**Keywords** Vascular Malformations; Targeted Therapies; Mosaicism; PIK3CA; RAS
**Subject Categories** Cardiovascular System; Vascular Biology & Angiogenesis

## Introduction to vascular malformations

Vascular anomalies encompass a wide group of developmental disorders affecting the vasculature. According to the 2025 International Society for the Study of Vascular Anomalies (ISSVA) classification, they are divided into vascular tumors and vascular malformations (VMs) (Goldenberg et al, 2025) (Classification | International Society for the Study of Vascular Anomalies). Vascular tumors are characterized by the abnormal proliferation of endothelial cells (ECs), which form the inner layer of blood and lymphatic vessels. In contrast, VMs result from inborn errors in vascular morphogenesis, leading to networks of structurally abnormal blood and/or lymphatic vessels, with minimal EC proliferation (Mulliken and Glowacki, 1982; Van Damme et al, 2020). Depending on the associated phenotype, VMs are further classified as simple, combined (when several types of vessels are involved), or complex when associated with nonvascular lesions (such as phosphatidylinositol-4,5-bisphosphate 3-kinase catalytic subunit alpha [*PIK3CA*]-related overgrowth syndromes, PROS). VMs are divided into fast-flow (arteriovenous malformations, AVMs) and slow-flow (lymphatic, capillary and venous) malformations, with different clinical and prognostic implications (Goldenberg et al, 2025) (Classification | International Society for the Study of Vascular Anomalies).

Historically, clinical classification has informed treatment decisions and offered prognostic guidance. However, misdiagnosis is extremely frequent in vascular anomalies: the term 'hemangioma' (a type of vascular tumor) has been shown to be misused in 71% of cases (Hassanein et al, 2011).

The prevalence of VMs is estimated around 1 in 1000 live births with an annual incidence of 1 per 10,000 (Penington et al, 2023; Ryu et al, 2023), though these figures are likely imprecise due to the absence of prospective registries. Venous malformations, the most common subtype, are found in 4.5 out of 10,000, while lymphatic malformations account for 3.5 out of 10,000. In contrast, AVMs only account for 1 case in 10,000, but they are associated with the highest mortality rates (Penington et al, 2023; Ryu et al, 2023).

Unlike some vascular tumors, VMs do not regress spontaneously and are frequently incurable. Complications depend on the type of lesion and include oozing or bleeding, venous thrombosis, sepsis, pain, impaired quality of life, and can be life-threatening. Given their chronic and progressive nature, these conditions necessitate lifelong follow-up in specialized tertiary centers, where access to expert multidisciplinary teams—including physicians, radiologists, surgeons, and geneticists—is essential yet often limited (Trenor Iii and Adams, 2020; Iacobas et al, 2022).

Recent advances in genetics have significantly advanced our understanding of VMs pathophysiology, by showing that oncogenic mutations are shared across different malformation subtypes. Though the pathophysiology of these complex disorders is still challenging, care for patients with VMs has drastically changed over the last few years. Genetic diagnosis is now routinely implemented, and oncologic targeted therapies are being repurposed with promising efficacy.

[1]INSERM U1151, Institut Necker-Enfants Malades, Paris, France. [2]Université Paris Cité, Paris, France. [3]Unité de Médecine Translationnelle et Thérapies Ciblées, Hôpital Necker-Enfants Malades, AP-HP, Paris, France. [4]Centre d'Investigation Clinique, Hôpital Necker-Enfants Malades, AP-HP, Paris, France. [5]These authors contributed equally: Gabriel Morin, Ilaria Galasso. ✉E-mail: guillaume.canaud@inserm.fr

**Glossary**

| | | | |
|---|---|---|---|
| Endothelial cells | Cells forming the inner monolayer of blood/lymphatic vessels, in direct contact with blood/lymph. | | endothelial proliferation. |
| Vascular anomalies | A group of rare disorders that gathers vascular tumors and vascular malformations. | Fast-flow vascular malformations | Vascular malformations involving fast-flow vessels, such as arteriovenous malformations. |
| Vascular tumors | Benign or malignant endothelial cell-derived tumors characterized by increased proliferation. | Slow-flow vascular malformations | Vascular malformations involving slow-flow vessels, including capillaries, veins and lymphatics. |
| Vascular malformations | Developmental anomalies of blood and lymphatic vessels, resulting in abnormal angio-architecture without markedly increased | Paradominance | A non-Mendelian mode of inheritance where disease manifests only after a post-zygotic second hit in individuals carrying a heterozygous germline variant. |

Because the molecular and therapeutic landscape of VMs is constantly evolving, this review aims to summarize the main genetic variants identified in VMs, current evidence on effective targeted therapies, and emerging treatments in development.

# Genetic bases of vascular malformations

Progress in molecular biology at the end of the 20th century has reshaped our knowledge of VMs. The first genetic anomalies associated with VMs were identified through studies of families with hereditary hemorrhagic telangiectasia (HHT), an autosomal dominant AVM syndrome. These investigations resulted in the identification of germline variants in *ENG*, the gene encoding endoglin (McAllister et al, 1994; Queisser et al, 2021). However, germline testing in patients with sporadic VMs frequently returns negative results (Borst et al, 2020). Moreover, familial VMs follow a paradominant inheritance pattern, where inherited variants frequently combine with somatic second-hit mutations to drive the phenotype (Seront et al, 1993; Brouillard and Vikkula, 2007). Consequently, germline variants usually lead to multifocal asymmetrical malformations depending on localized additional events, as shown in various familial VM syndromes (Limaye et al, 2009a; Revencu et al, 2013; Amyere et al, 2013; Macmurdo et al, 2016; Hill et al, 2021; DeBose-Scarlett et al, 2024; Castillo et al, 2025). The identification of such somatic second hits in families with germline VM syndromes at the beginning of the 21st century then paved the way for somatic genetic testing in sporadic VMs (Brouillard et al, 2002). In the following years, a growing number of genes involved in EC homeostasis have been reported as VM drivers.

## Somatic mosaicism in vascular malformations

In the 1970s, the concept of somatic mosaicism was proposed as the underlying mechanism of segmental skin disorders (Happle, 1978). Mosaicism is defined as the existence of at least two genetically distinct cell populations in a single individual. In sporadic VMs, postzygotic mutations have been reported in various genes regulating EC function, resulting in segmental phenotypes and reinforcing the importance of targeted biopsies of affected tissue for accurate genetic diagnosis. The first somatic variants associated with sporadic VMs were found in *TEK* (Tyrosine kinase with immunoglobulin and EGF homology domains), in patients with venous malformations (Limaye et al, 2009b). *TEK* encodes for TIE2, a receptor-tyrosine kinase (RTK) involved in EC signaling (Fig. 1). Numerous other genes were then reported as causative in

various types of VMs, such as AK strain Transforming gene (*AKT*, in Proteus syndrome), *PIK3CA* (in PROS) or Kirstein RAt Sarcoma viral oncogene homolog (*KRAS*, in brain AVMs) (Lindhurst et al, 2011; Kurek et al, 2012; Nikolaev et al, 2018). Interestingly, all of these genes had already been reported as driver oncogenes in malignant tumors (McCoy et al, 1983; Samuels, 2004; Carpten et al, 2007). Though VMs are nonmalignant conditions, shared genetic background with cancer has recently motivated repurposing of oncology drugs with promising efficacy.

## Molecular mechanisms involved in sporadic vascular malformations

Over the last decades, genetic discoveries in VMs have revealed several critical signaling pathways in ECs. These pathways integrate extracellular growth signals regulating proliferation, survival, apoptosis, differentiation and metabolism. They include the RAS-Mitogen Associated Protein Kinase (MAPK) pathway, the Phosphoinositide 3-Kinase (PI3K) pathway, the Gα pathway, the Transforming Growth Factor β (TGF-β) pathway, and the Cerebral Cavernous Malformation (CCM) pathway.

Loss-of-function (LoF) variants are more common in germline diseases, often involving a mosaic second hit, while gain-of-function (GoF) variants are more prevalent in sporadic disorders. Most VMs are sporadic and predominantly caused by somatic pathogenic variants in genes involved in these pathways, leading to dysregulated angiogenesis. Interestingly, clinical manifestations depend on the affected pathway. Indeed, GoF variants involving *PIK3CA* (which is involved in Vascular Endothelial Growth Factor (VEGF) Receptor [VEGFR] signaling) are prone to induce slow-flow VMs but almost never AVMs (Canaud et al, 2021). On the other hand, AVM variants most frequently involve genes of the RAS-MAPK pathway (Nikolaev et al, 2018; Al-Olabi et al, 2018; Li et al, 2018; Hong et al, 2019; Konczyk et al, 2019; Schmidt et al, 2024). Consequently, the higher prevalence of somatic *PIK3CA* variants compared to MAPK pathway gene variants likely accounts for the greater frequency of slow-flow VMs relative to AVMs (Penington et al, 2023; Ryu et al, 2023).

## Endothelial signaling in vascular malformations

### TIE2-angiopoietin signaling and the PI3Kα pathway

Most slow-flow VMs are sporadic and caused by pathogenic variants in genes involved in the PI3Kα and the TIE2-angiopoietin pathways. Among them, venous and lymphatic malformations (LMs, Fig. 2) are the most common (Behravesh et al, 2016; Penington et al, 2023). Slow-flow VMs can be isolated or part of complex syndromes such as PROS,

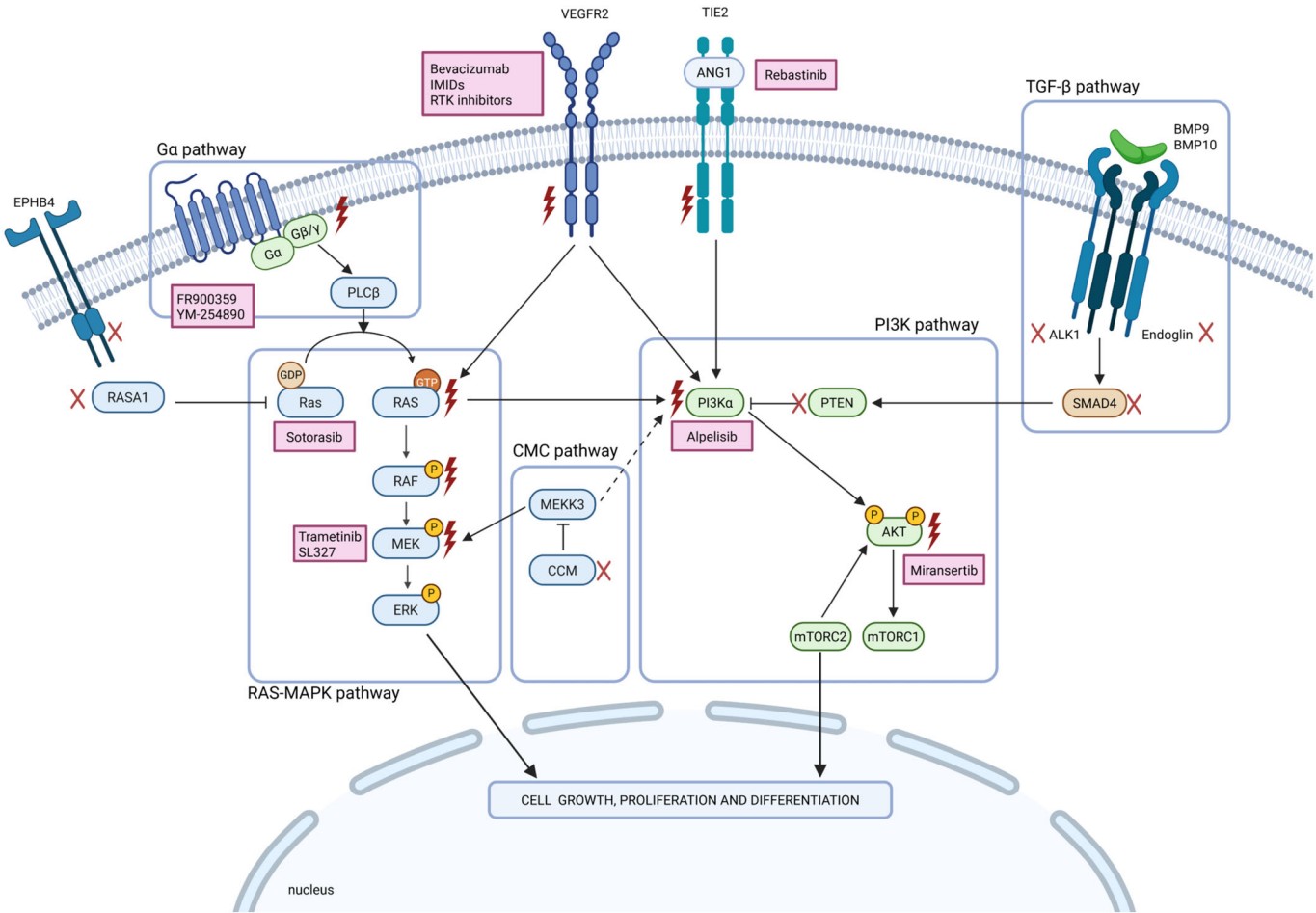

**Figure 1.  Signaling pathways involved in vascular malformations.**

Pharmacological inhibitors are indicated in boxes close to their respective targets. Crosses and lightning bolts denote components with loss-of-function and gain-of-function variants reported in vascular malformations, respectively. Sharp arrows indicate direct (solid line) or indirect (dashed line) activation. Blunt arrows show inhibition. ALK1 activin receptor-like kinase 1, ANG angiopoietin, AKT AK strain transforming, BMP9/10 bone morphogenetic protein 9/10, CCM cerebral cavernous malformation complex, EPHB4 Ephrin B4, ERK extracellular signal regulated kinase, IMiDs immunomodulatory drugs, MAPK mitogen-associated protein kinase, MEK mitogen-activated protein kinase kinase, MEKK3 mitogen-activated protein kinase kinase kinase 3, mTORC1/2 mammalian target of rapamycin complex 1/2, PI3Kα phosphoinositide 3 kinase α, PLCβ phospholipase Cβ, PTEN phosphatase and tensin homolog, RAF rapidly accelerated fibrosarcoma, RAS rat sarcoma viral oncogene homolog, RASA1 RAS p21 protein activator 1, RTK receptor-tyrosine kinase, SMAD4 mothers against decapentaplegic homolog 4, TGF-β transforming growth factor β, TIE2 tyrosine kinase with immunoglobulin and EGF homology domains, VEGFR vascular endothelial growth factor receptor.Created in BioRender. Galassi, I. (2025) https://BioRender.com/o21g614. Figure created with BioRender.com.

which involves asymmetrical overgrowth of surrounding soft tissues and/or bones in addition to VMs.

The TIE2 receptor (encoded by *TEK*) is expressed at the surface of ECs and binds angiopoietins (ANG), a family of endothelial growth factors (Fig. 1). Angiopoietin 1 (ANG1) is secreted by perivascular cells such as vascular smooth muscle cells (VSMCs) and promotes vascular maturation and stability (Augustin et al, 2009; Thomas and Augustin, 2009), while angiopoietin 2 (ANG2) signals in ECs in an autocrine fashion (Yuan et al, 2009; Korhonen et al, 2016). Activation of TIE2 initiates the PI3Kα pathway, which is critical for vascular maturation, remodeling, and angiogenesis (Seront et al, 1993).

The PI3Kα pathway is further activated by VEGFRs which regulate angiogenesis: VEGFR2 binds VEGF-A in blood ECs, while VEGFR3 binds VEGF-C in lymphatic ECs (Shibuya, 2011; Kuonqui et al, 2025). PI3Kα is a heterodimer made of p110α (catalytic subunit, encoded by *PIK3CA*) and p85 (regulatory subunit, encoded by *PIK3R1*), that plays a central role in intracellular signaling by phosphorylating membrane lipids called phosphoinositides (Fruman et al, 2017). Once activated by ligand-bound RTKs, PI3Kα phosphorylates phosphatidyl-innositol-4,5-biphosphate (PIP2) into phosphatidyl-innositol-3,4,5-triphosphate (PIP3). PIP3 activates Phosphoinositide-Dependent Kinase 1 (PDK1) which, along with mammalian Target Of Rapamycin 2 (mTORC2), activates the serine/threonine kinase AKT. In turn, AKT induces a proliferative and anti-apoptotic program by inhibiting the Forkhead box O (FoxO) family of proteins and activating mammalian target of rapamycin (mTOR) complex 1 (mTORC1) (Vanhaeseb-roeck et al, 2010). The tumor suppressor phosphatase and tensin homolog (PTEN) is the main inhibitor of the pathway, counteracting PI3Kα activity by converting PIP3 back to PIP2 (Chalhoub and Baker, 2009).

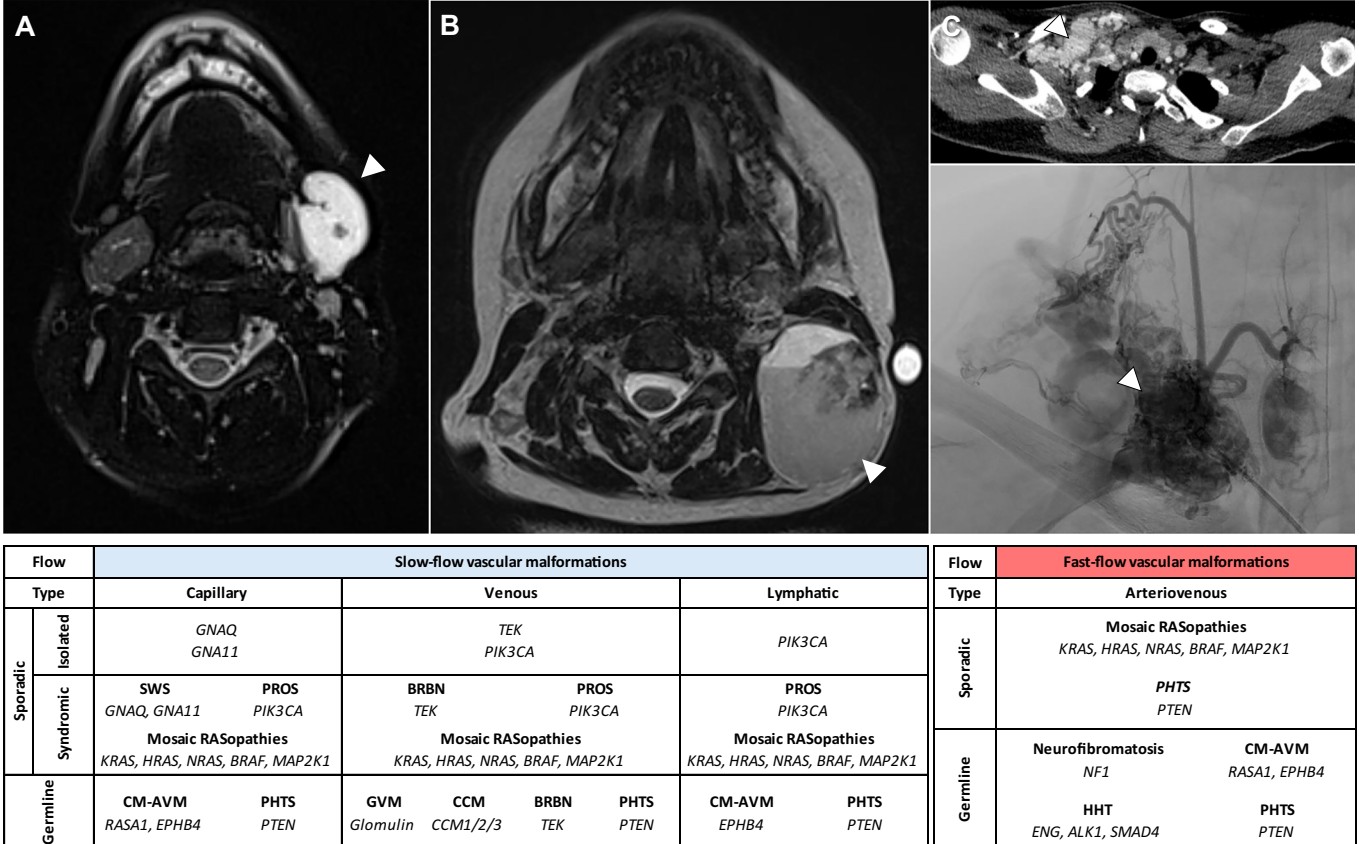

| Flow | | Slow-flow vascular malformations | | | | Flow | Fast-flow vascular malformations |
|---|---|---|---|---|---|---|---|
| Type | | Capillary | Venous | | Lymphatic | Type | Arteriovenous |
| **Sporadic** | Isolated | *GNAQ* *GNA11* | *TEK* *PIK3CA* | | *PIK3CA* | **Sporadic** | Mosaic RASopathies *KRAS, HRAS, NRAS, BRAF, MAP2K1* |
| | Syndromic | **SWS** *GNAQ, GNA11*   **PROS** *PIK3CA* | **BRBN** *TEK*   **PROS** *PIK3CA* | | **PROS** *PIK3CA* | | *PHTS* *PTEN* |
| | | Mosaic RASopathies *KRAS, HRAS, NRAS, BRAF, MAP2K1* | Mosaic RASopathies *KRAS, HRAS, NRAS, BRAF, MAP2K1* | | Mosaic RASopathies *KRAS, HRAS, NRAS, BRAF, MAP2K1* | | |
| **Germline** | | **CM-AVM** *RASA1, EPHB4*   **PHTS** *PTEN* | **GVM** *Glomulin*   **CCM** *CCM1/2/3*   **BRBN** *TEK*   **PHTS** *PTEN* | | **CM-AVM** *EPHB4*   **PHTS** *PTEN* | **Germline** | **Neurofibromatosis** *NF1*   **CM-AVM** *RASA1, EPHB4* |
| | | | | | | | **HHT** *ENG, ALK1, SMAD4*   **PHTS** *PTEN* |

**Figure 2.   Recurrent genetic variants found in vascular malformations.**

(**A**) Venous malformation of the left face (arrowhead, MRI scan). (**B**) Cervical lymphatic malformation (arrowhead, MRI scan). (**C**) Extracranial supraclavicular arteriovenous malformation (arrowheads; upper panel, angio-CT scan; lower panel, angiography). **Table**. Most common variant genes found in vascular malformations (non-exhaustive list). BRBN blue rubber bleb nevus syndrome, CCM cerebral cavernous malformation, CM-AVM capillary malformation-arteriovenous malformation syndrome, HHT hereditary hemorrhagic telangiectasia, MCAP megalencephaly-capillary malformation-polymicrogyria syndrome, PHTS PTEN-hamartoma tumor syndrome, PROS PIK3CA-related overgrowth spectrum. Images courtesy of Dr Antoine Fraissenon (Service de Radiologie Pédiatrique, Hospices Civils de Lyon, Lyon, France).

In 1996, germline GoF *TEK* variants were identified as the cause of cutaneous-mucosal venous malformations, a rare autosomal dominant VM syndrome (Vikkula et al, 1996). Further studies revealed somatic GoF *TEK* variants (mainly L914F) in ~50% sporadic venous malformations, and in *PIK3CA* (mainly hotspot variants E542K, E545K, and H1047R) in 20% (Limaye et al, 2009b; Soblet et al, 2013; Limaye et al, 2015; Castel et al, 2016; Castillo et al, 2016). Interestingly, *TEK* and *PIK3CA* variants were found to be mutually exclusive. Conversely in sporadic LMs, somatic *PIK3CA* variants are more prevalent and account for ~80% cases (Luks et al, 2015; Osborn et al, 2015). Less frequently, somatic *PIK3R1* variants have also been reported in patients with phenotypes overlapping with PROS, including LMs and venous malformations (Cottrell et al, 2021; Schönewolf-Greulich et al, 2022; Morin et al, 2025b). Additionally, LoF *PTEN* variants (both germline and mosaic) have been reported to cause PTEN hamartoma tumor syndrome (PHTS), which may involve VMs (Liaw et al, 1997; Salo-Mullen et al, 2014; Hendricks et al, 2022; Castillo et al, 2025). Interestingly, *PTEN* variants are frequently associated with AVMs (up to 86% of *PTEN*-related VAs), while *PIK3CA* variants are not, suggesting *PTEN*-related AVMs may result from PI3Kα-independent mechanisms (Tan et al, 2007).

Dysregulation of the PI3Kα pathway has morphological and functional consequences in ECs. GoF variants in *TEK* and *PIK3CA* lead to excessive PI3Kα, AKT and mTOR signaling, driving senescence, increased vessel permeability, altered arteriovenous identity, and aberrant angiogenic potential (Bloom et al, 2023; Sabata et al, 2025). In vitro studies showed that human umbilical vein ECs (HUVECs) expressing *PIK3CA* or *TEK* GoF variants become dysmorphic and produce a defective angiogenic secretome (Castel et al, 2016). In mouse models, constitutive activation of *PIK3CA* leads to embryonic lethality due to disrupted vasculogenesis and hematopoiesis (Hare et al, 2015). Moreover, endothelial-specific *PIK3CA* GoF variants result in abnormal retina vessels with reduced pericyte coverage and anarchic EC proliferation (Castillo et al, 2016; Kobialka et al, 2022).

Additionally, recent work proposed that a PI3K-ANG2-TIE2 amplification loop may participate in the growth of venous malformations. In *PIK3CA*-mutant ECs, AKT-mediated FOXO1 inhibition lowers ANG2 synthesis. The resulting ANG1/ANG2 imbalance may account for the increased TIE2 phosphorylation seen in these lesions. This model is further supported by the finding that TIE2 inhibition—despite acting upstream of PIK3CA—reduces VM growth in vivo (Kraft et al, 2025).

Finally, activation of the PI3Kα pathway in the endothelium leads to a coagulation disorder called localized intravascular coagulation (LIC) in both animal models and patients with venous malformations (Castel et al, 2016; Castillo et al, 2016; Sterba et al, 2023; Zerbib et al, 2024). LIC, marked by elevated D-dimer levels, reduced platelet counts, and decreased fibrinogen, predisposes to thromboembolic events and may necessitate anticoagulant therapy (Dompmartin et al, 2008). Likewise in lymphatic ECs, *PIK3CA* GoF variants result in diffuse LMs in vivo, with lymphatic leakage and excessive branching (Rodriguez-Laguna et al, 2019). Similar to the amplification loop described in venous malformations, VEGF-C promotes the growth of *PIK3CA*-mutated LMs by further activating the PI3K pathway (Martinez-Corral et al, 2020). Additionally, endothelial-macrophage crosstalk may reinforce this process: *PIK3CA*-mutated lymphatic ECs were shown to secrete chemokines that recruit macrophages, which in turn produce VEGF-C and sustain lymphangiogenesis (Petkova et al, 2023).

The PI3K pathway is therefore a crucial therapeutic target in slow-flow VMs.

### The CCM pathway

Cerebral cavernous malformations (CCMs) are venous malformations of the brain at risk of bleeding, resulting in epilepsy, focal neurological deficits, or headaches (Snellings et al, 2021). Familial CCM syndromes result from germline mutations in one of the three *CCM* genes (*CCM1, 2* and *3*) which encode components of the CCM complex. This complex acts as an inhibitor of MEKK3 (encoded by *MAP3K3*), involved in the MAPK pathway (Fig. 1) (Zhou et al, 2016). LoF of CCM complex results in increased MEKK3 signaling and activation of the mTOR pathway (Zhou et al, 2016; Ren et al, 2021).

CCMs development often follows a two-step mechanism involving both the CCM-MAP3K3 axis and the PI3K pathway. In familial CCMs, germline mutations in the *CCM* genes are frequently associated with an additional postzygotic variant in *PIK3CA* (Ren et al, 2021). Similarly, sporadic CCMs may involve postzygotic variants in *PIK3CA* and additional variants in the *CCM* genes or in *MAP3K3*. Interestingly, sporadic CCMs often arise near developmental venous anomalies (DVA), a type of brain venous malformation caused by somatic *PIK3CA* variants, which may serve as a primer for CCMs (Petersen et al, 2010; Snellings et al, 2022). As both *PIK3CA* GoF and *CCM* LoF converge towards the activation of mTORC1, they probably synergistically contribute to CCM growth (Ren et al, 2021).

### The RAS-MAPK pathway

Because of their overlapping phenotypes, disorders caused by somatic variants in the RAS-MAPK pathway have been collectively gathered as mosaic RASopathies. While variants in genes of the PI3K pathway usually result in slow-flow VMs, the RAS-MAPK pathway is frequently involved in AVMs. Indeed, Schmidt et al reported that up to 92.3% of extracranial VMs caused by mosaic mutations in the RAS-MAPK pathway are AVMs, either isolated or combined (Schmidt et al, 2024). Mosaic RASopathies associate with high morbidity, as brain AVMs are a leading cause of nontraumatic brain hemorrhage in children and young adults (Boulouis et al, 2019; Tatlisumak et al, 2018).

The RAS-MAPK pathway is a central regulator of cell proliferation. RAS proteins are small GTPases that act like molecular switches, encoded by three different genes, *KRAS*, Harvey-RAS (*HRAS*) and Neuroblastoma-RAS (*NRAS*). Activated RTKs, such as VEGFRs, recruit adaptor proteins known as guanine nucleotide exchange factors (GEFs), including Son-Of-Seventy 1 (*SOS1*). GEFs activate RAS by promoting the exchange of GDP by GTP. Conversely, GTPase-activating proteins (GAP) such as Neurofibromatosis 1 (*NF1*) or RAS p21 protein activator 1 (*RASA1*), activated by the Ephrin B4 (*EPHB4*) receptor, inactivate RAS by increasing its GTP hydrolysis rate (Downward, 2003; Simanshu et al, 2017). Once activated, RAS engages multiple effectors by binding to their RAS-binding domains (RBDs), including RAF, which initiates the MAPK pathway, and PI3Kα (Fig. 1) (Downward, 2003). RAF proteins (ARAF, BRAF and c-RAF1) phosphorylate MEK1/2, which in turn activates ERK1/2. ERK finally phosphorylates several cytosolic and nuclear targets to promote cell cycle progression (Downward, 2003).

Most frequently in AVMs, mosaic GoF variants are found in *RAS*, *BRAF*, *MAP2K1* (encoding for MEK1), *ARAF* or *SOS1* (Fig. 2) (Jauhiainen et al, 2018; Al-Olabi et al, 2018; Hong et al, 2019; Schmidt et al, 2024). In extracranial AVMs, *RAS* variants tend to associate with higher severity and more frequent progression after treatment, compared to variants in other genes of the MAPK pathway (Schmidt et al, 2024). Additionally, LoF variants are found in germline conditions such as capillary malformation-AVM syndrome (CM-AVM, caused by *RASA1* or *EPHB4* variants), neurofibromatosis (*NF1*) and Noonan syndrome (*PTPN11*), and have been reported as mosaic events in sporadic VMs (Fig. 2) (Al-Olabi et al, 2018; Wooderchak-Donahue et al, 2019; Revencu et al, 2020; Carli et al, 2022; Schmidt et al, 2024).

Apart from AVMs, variants in the RAS-MAPK pathway may occasionally cause complex lymphatic anomalies (CLAs) and capillary or venous malformations. CLAs gather four exceedingly rare syndromes involving the lymphatic system with severe local and systemic complications. They may involve diffuse visceral LMs (generalized lymphatic anomalies, GLA), prominent osseous LMs (Gorham-Stout disease, GSD), proliferative lymphatic ECs with diffuse intravascular coagulation (DIC) and high mortality rates (kaposiform lymphangiomatosis, KLA), or central lymphatic conduct malformations causing serous effusion and massive edema (central conducting lymphatic anomalies, CCLA) (Barclay et al, 2019; Ricci and Iacobas, 2022; Ramwani et al, 2023; Torales et al, 2024). Variants in *NRAS*, *HRAS*, *KRAS*, *ARAF*, *BRAF* and *MAP2K1* have been reported in CLAs (Fig. 2) (Nozawa et al, 2020; Liu et al, 2022b; Sheppard et al, 2023; Li et al, 2023a; Torales et al, 2024).

KLA exhibits the poorest prognosis among CLAs, with high mortality rates (Eng et al, 2018; Perez-Atayde et al, 2022). Similar to the Kasabach-Merritt phenomenon (KMP)—a severe localized thrombotic process leading to consumptive coagulopathy in vascular tumors such as kaposiform hemangioendothelioma—KLA often presents with a KMP-like syndrome resulting in life-threatening DIC (Kelly, 2010; Perez-Atayde et al, 2022; Servattalab and Grenier, 2025). KLA is almost exclusively driven by the Q61R variant of *NRAS*, suggesting that lymphatic homeostasis relies heavily on this isoform (Manevitz-Mendelson et al, 2018; Barclay et al, 2019; Ozeki et al, 2019; Li et al, 2023a; Ramwani et al, 2023).

Finally, RAS variants also account for a minority of capillary and venous malformations with asymmetrical overgrowth, often alongside AVMs (Fig. 2) (Al-Olabi et al, 2018; Schmidt et al, 2024).

### The TGF-β pathway

Variants involving the TGF-β pathway are found in germline autosomal dominant disorder HHT (Rendu-Osler's disease). HHT

is associated with multifocal AVMs and high-output cardiac failure owing to severe shunting (Shovlin, 2010).

In ECs, Bone Morphogenetic Protein 9 (BMP9) activates the Activin-like kinase 1 (ALK1) receptor and its coreceptor ENG (Fig. 1). The ALK1/ENG complex indirectly induces Mothers against decapentaplegic homolog 4 (SMAD4) phosphorylation. SMAD4 then translocates to the nucleus and promotes the transcription of several target genes, including *VEGFR1* (a decoy receptor competing with VEGFR2 for binding VEGF) and *PTEN*, decreasing the activity of the VEGFR and the PI3K pathway (Tabosh et al, 2024). Therefore, LoF variants of the TGF-β pathway result in increased PI3K signaling. Finally, impaired TGF-β signaling leads to dysregulated angiogenesis and abnormal VSMC differentiation (Deng et al, 2024; Chen et al, 2023b).

In HHT, germline mutations were reported in *BMP9*, *ALK1*, *ENG* and *SMAD4* (Fig. 2) (McAllister et al, 1994; Johnson et al, 1996; Gallione et al, 2006; Wooderchak-Donahue et al, 2013). Interestingly, somatic second hits resulting in loss of heterozygosity have been found in HHT-related AVMs (Snellings et al, 2019; DeBose-Scarlett et al, 2025).

### The Gα pathway

Other VMs such as capillary malformations may result from anomalies in the Gα pathway. They can be isolated or part of Sturge-Weber syndrome (SWS), a congenital disorder involving facial and leptomeningeal capillary malformations and neurological complications such as epilepsy (Shirley et al, 2013).

Somatic GoF variants have been reported in genes encoding G proteins in capillary malformations (Shirley et al, 2013; Couto et al, 2017; Polubothu et al, 2020). Gαq and Gα11, encoded by guanine nucleotide-binding protein subunit alpha q (*GNAQ*) and 11 (*GNA11*), participate in G protein-coupled receptors (GPCR) signaling. They activate phospholipase C-β (PLCβ), which in turn activates the RAS-MAPK pathway (Figs. 1 and 2). Two mutational hotspots are found in *GNAQ* and *GNA11* at codons R183 and Q209 (Shirley et al, 2013; Ayturk et al, 2016; Galeffi et al, 2022). The most common variant in VMs, *GNAQ* R183Q, accounts for 90% of non-syndromic cases and SWS, while *GNAQ*-negative samples frequently harbor the *GNA11* R183C variant (Shirley et al, 2013; Polubothu et al, 2020). *GNAQ* R183Q reduces Gαq affinity for GDP and results in sustained activation. Consequently, mutant *GNAQ* dysregulates the MAPK pathway which may impair ECs differentiation and flow response, leading to capillary malformations (Shirley et al, 2013). Interestingly, *GNAQ* variants are also found in vascular tumors such as congenital hemangiomas (Ayturk et al, 2016). However, tumor-associated *GNAQ* variants seem to more strongly affect the Gα pathway. Indeed, the most common variant in hemangiomas, Q209L, activates Gα signaling more strongly than R183Q, which is predominant in VMs (Ayturk et al, 2016; Martins et al, 2017; Galeffi et al, 2022). Thus, the distinct prevalence of *GNAQ* variants in vascular tumors and malformations likely reflects their unique functional effects.

# Targeted therapies for vascular malformations

Treatment of VMs usually requires a multimodal approach, including surgery, sclerotherapy, or endovascular embolization when necessary. However, therapeutic options remain limited when interventional procedures are ineffective or unfeasible. In recent years, genetic sequencing has gained theragnostic significance, helping to identify targeted therapies tailored to specific causal mutations. Many drugs used to treat VMs were originally developed for oncology or organ transplantation. Unlike oncology, which often employs combination therapies, VM treatments typically target a single pathway. Due to the rarity of VMs, clinical trials are limited and most studies are retrospective. This section reviews the latest therapeutic advances in VMs.

## Inhibitors of the PI3K pathway

Variants in the PI3K pathway are the most common in slow-flow VMs, and several oncology drugs are now being repurposed for these disorders.

### PI3Kα inhibitors

Alpelisib is a competitive selective PI3Kα inhibitor (Fig. 1) originally developed for the treatment of *PIK3CA*-mutated advanced breast cancer (André et al, 2019; Center for Drug Evaluation and Research, 2019). Alpelisib has also been extensively assessed in *PIK3CA*-related VMs. It showed efficacy at restoring cell morphology in *PIK3CA*[H1047R]-mutant HUVECs and improved vascular morphogenesis and coagulation markers in vivo (Limaye et al, 2015; Castel et al, 2016). Interestingly, it showed better efficacy at improving *PIK3CA*-related VMs and survival in vivo, compared to sirolimus (Venot et al, 2018). Subsequent clinical studies confirmed alpelisib efficacy in PROS patients, including infants (Venot et al, 2018; Pagliazzi et al, 2021; Morin et al, 2022; Garreta Fontelles et al, 2022; Schönewolf-Greulich et al, 2022; Canaud et al, 2023; Cossio et al, 2024). Alpelisib showed effectiveness across various types of *PIK3CA*-driven VMs, reducing target lesion volumes in LMs and venous/capillary VMs by 48% and 33%, respectively (Delestre et al, 2021; Sterba et al, 2023; Zerbib et al, 2024). Interestingly, it also showed effectiveness in VMs caused by variants in genes encoding regulators of PI3Kα such as *TEK* (Remy et al, 2022; Sterba et al, 2023; Zerbib et al, 2024) and *PIK3R1* (Schönewolf-Greulich et al, 2022; Morin et al, 2025b). The EPIK-P1 clinical trial (NCT04285723) eventually reported a decrease in target lesion volume in 74.2% of PROS patients treated with alpelisib for 6 months, with a mean reduction of 13.7%, and led to alpelisib approval by the United States Food and Drug Administration (FDA) (Center for Drug Evaluation and Research, 2022; Canaud et al, 2023). However, the EPIK-P2 trial did not reach the anticipated 20% reduction in lesion volume, likely due in part to the reduced alpelisib dose administered to adults (125 mg instead of the FDA-approved 250 mg) (Canaud et al, 2024). Additionally, the EPIK-L1 phase II/III trial (NCT05948943, Table 1) is currently assessing alpelisib in *PIK3CA*-related LMs. Finally, the ongoing SESAM study evaluates alpelisib in megalencephaly-capillary malformation-polymicrogyria syndrome (MCAP, a PROS subtype), following reports of neurological and MRI improvements in treated patients (Venot et al, 2018; Morin et al, 2022; Luu et al, 2024).

PI3Kα is also involved in VMs caused by variants in the RAS-MAPK pathway. In *HRAS*[G12V]-mutant ECs, activation of the PI3Kα pathway by RAS results in abnormal morphogenesis (Bajaj et al, 2010). Moreover, ECs expressing a modified *HRAS*[G12V] variant unable to bind PI3Kα show decreased proliferation rates, while

**Table 1. Clinical trials dedicated to vascular malformations.**

| Vascular malformation types | Target | Drug | Main objective | Phase | Clinical trial ID number |
|---|---|---|---|---|---|
| Lymphatic Malformations | mTOR | Sirolimus | Safety and efficacy | 2–3 | NCT06673290 |
| Microcystic Lymphatic Malformations | mTOR | Sirolimus | Safety and efficacy | Not applicable | NCT06160739 |
| Cervico-facial Lymphatic Malformations | mTOR | Sirolimus | Safety and efficacy | 2 | NCT03243019 |
| Microcystic Lymphatic Malformations | mTOR | Topical sirolimus | Safety and efficacy | 2 | NCT04128722 |
| Lymphatic Malformations | PI3Kα | Alpelisib | Safety and efficacy | 2–3 | NCT05948943 |
| Brainstem Cavernous Malformations | mTOR | Sirolimus | Safety and efficacy | 2 | NCT06091332 |
| Slow-Flow Vascular Malformations | PI3Kα | Alpelisib | Safety and efficacy | 2 | NCT05983159 |
| MCAP | PI3Kα | Alpelisib | Safety and efficacy | 2 | NCT05577754 |
| Arteriovenous Malformations | mTOR | Sirolimus | Safety and efficacy | 2 | NCT02042326 |
| Arteriovenous Malformations | MEK | Trametinib | Safety and efficacy | 2 | NCT06098872 |
| Arteriovenous Malformations | MEK | Trametinib | Safety and efficacy | 2 | NCT04258046 |
| Arteriovenous Malformations | MEK | Trametinib | Safety and efficacy | 2 | EudraCT 2019-003573-26 |
| Extracranial Arteriovenous Malformations | MEK | Cobimetinib | Safety and efficacy | 2 | NCT05125471 |
| Fast-Flow Vascular Malformations | MEK | Mirdametinib | Safety and efficacy | 2 | NCT05983159 |
| Hereditary Hemorrhagic Telangiectasia | AKT | VAD044 | Safety and efficacy | 1 | NCT05406362 |
| Hereditary Hemorrhagic Telangiectasia | VEGFR | Pazopanib | Efficacy | 1–2 | NCT03850730 |
| Hereditary Hemorrhagic Telangiectasia | VEGFR | Pazopanib | Safety and efficacy | 2–3 | NCT03850964 |

MCAP megalencephaly-capillary malformation polymicrogyria syndrome.

treatment with PI3K inhibitors improves both *HRAS*-mutant cells morphogenesis and proliferation (Bajaj et al, 2010; Li et al, 2018). However, alpelisib did not improve survival nor brain VMs in an endothelial *HRAS^{G12V}*-mutant mouse model (Li et al, 2018). Likewise, Fish et al showed that LY294002, a pan-PI3K inhibitor, did not improve AVMs in a *KRAS^{G12V}*-mutant zebrafish model (Fish et al, 2020).

Nevertheless, LY294002 improved retinal vasculature hyperplasia in a germline *ALK1*-mutant mouse model of HHT. These results were further confirmed in *ALK1*-mutant mice harboring a germline heterozygous kinase-dead *Pik3ca* (Alsina-Sanchís et al, 2018). So far, no clinical study supports the use of alpelisib in AVMs.

PI3Kα inhibitors impact the insulin pathway and can cause dose-dependent hyperglycemia or diabetes that may require the use of antidiabetic drugs (André et al, 2019; Sterba et al, 2023; Canaud et al, 2023; Remy et al, 2025). Other frequent AEs include rash, aphthous ulcers and diarrhea, that usually resolve after treatment discontinuation (Wang et al, 2020; Canaud et al, 2023). In the SOLAR phase III study assessing alpelisib (300 mg/d) in 284 patients with *PIK3CA*-mutated, hormone receptor-positive breast cancer, hyperglycemia (64% patients), diarrhea, nausea, decreased appetite and rash were the most frequent all-grade AEs (André et al, 2019). Similar AEs had been previously reported in an oncology dose escalation study (up to 450 mg/d alpelisib) and in a phase 1b study, where hyperglycemia and rash were the most common dose-dependent, alpelisib-related AEs. Hyperglycemia was manageable by dose interruption or metformin in most patients, while rash occurred within 2 weeks of treatment and was manageable by antihistamines and steroids (Juric et al, 2018, 2019).

Less is known about the long-term AEs of alpelisib, especially on growth and puberty. In the EPIK-P1 trial, drug-related AEs occurred in 38.6% of patients, notably hyperglycemia (27.8% of adults, 5.1% of children, grade ≤2) and aphthous ulcers (16.7% and 7.7%, respectively) (Canaud et al, 2023). In the EPIK-P2 trial, drug-related AEs occurred in 57.4% of adults receiving alpelisib (versus 44.4% in the placebo group) and 33.9% of children (versus 32.1%) (Canaud et al, 2024, 2). Though no warning has been issued by health authorities nor by most of the available literature so far (Sterba et al, 2023; Pagliazzi et al, 2021; Garreta Fontelles et al, 2022; Schönewolf-Greulich et al, 2022; Canaud et al, 2023; Kolitz et al, 2022), cases of growth delay were recently reported in pediatric patients treated with alpelisib (Cossio et al, 2024; Etingin et al, 2025; Triana et al, 2025). In a retrospective study, growth restriction was reported in up to 16.7% of patients receiving alpelisib, mostly in young children (<10 y) and in those receiving high weight-adjusted doses (3.6–4.8 mg/kg/day), suggesting a dose-dependent effects due to increased drug exposure (Cossio et al, 2024; Etingin et al, 2025; Triana et al, 2025). We did not observe such complications in the infants we treated with lower weight-adjusted doses (2.8 and 3.1 mg/kg/day), suggesting that growth delay may occur at higher alpelisib doses (Morin et al, 2022). As highlighted in a recent report, determining the optimal weight-adjusted dosage of alpelisib in the pediatric population based on pharmacokinetic data is therefore essential (Remy et al, 2025). Until long-term follow-up data from large therapeutic cohorts become available, pediatric patients receiving alpelisib should be closely monitored.

### AKT inhibitors

Since *PIK3CA*-mutant ECs rely heavily on AKT signaling, targeting AKT offers a promising approach for treating slow-flow VMs. Miransertib, a selective allosteric AKT inhibitor (Fig. 1), has shown effectiveness at preventing *PIK3CA*-driven VMs and improving already established lesions in vivo (Kobialka et al, 2022). In a

teenager with Proteus syndrome, miransertib led to a decrease in the volume of an ovarian tumor and complete resolution of a portal vein thrombosis (Leoni et al, 2019; Ours et al, 2021). It was also suggested to be effective and well-tolerated in two children with sirolimus-refractory PROS after a median duration of 22 months (Forde et al, 2021). However, in vivo data suggested that miransertib may be less effective than alpelisib in *PIK3CA*-related venous malformations (Zerbib et al, 2024). More recently, capivasertib, another AKT inhibitor, showed comparable efficacy as observed with sirolimus at reducing mTORC1 signaling and retinal vascular hyperplasia in a preclinical model of *PTEN*-related VMs (Castillo et al, 2025). Finally, AKT inhibitors have recently showed preclinical efficacy in a zebrafish model of KLA, suggesting that disorders involving both the PI3K and MAPK pathways might be accessible to similar therapies (Bassi et al, 2025).

AEs reported with AKT inhibitors include skin rash and nausea (Leoni et al, 2019). Additionally, miransertib's lack of isoform selectivity may cause glucose metabolism disorders similar to those seen with alpelisib.

### Rapalogs

Sirolimus, an mTOR inhibitor (Fig. 1), was the first and most extensively studied targeted therapy in VMs (Blatt et al, 2010). It was originally approved by the US FDA as an immunosuppressant for kidney transplantation. Later, evidence of activation of the mTOR pathway in VMs encouraged its off-label use (Hammill et al, 2011).

Sirolimus inhibits proliferative signaling and protein synthesis induced by mTOR, exerting antitumoral and antiangiogenic effects. Sirolimus plasma levels can be monitored routinely, and its extensive use in oncology and organ transplant has provided valuable insights into its mechanism of action and AEs. Numerous retrospective studies have evaluated sirolimus in both slow-flow and fast-flow VMs with variable efficacy (Hammill et al, 2011; Vlahovic et al, 2015; Wang et al, 2015; Nadal et al, 2016). Preliminary results from the European multicentric Phase III Vascular Anomaly-Sirolimus-Europe (VASE) trial, conducted on 132 pediatric and adult patients with slow-flow VMs, showed a symptomatic improvement in 85% of patients after two years on sirolimus. However, no data was reported regarding VMs volume (Seront et al, 2023). Sirolimus also showed promising results in CLAs: VM volume decreased and coagulation parameters improved in 58% of 7 treated KLA cases in a combined study (Zhou et al, 2021). In a retrospective study of 44 KLA patients, 83% of the 24 patients treated with sirolimus showed sustained (>6 months) improvements in lesion size on imaging, DIC, and quality-of-life, although no improvement was observed in survival, with a median time from diagnosis to death of 2.2 years (Eng et al, 2018). Sirolimus also showed symptomatic efficacy in GLA and GSD, with improvements in quality of life and pleural effusion in roughly 75% of 18 affected patients, though only 28% showed improvement in lesions size on imaging (Ricci et al, 2019; Liang et al, 2020).

While sirolimus demonstrated effectiveness in LMs, it did not significantly improve sporadic venous malformations (Hammill et al, 2011; Maruani et al, 2021). However, in vivo studies suggested it could prevent the growth of cerebral cavernomas (Li et al, 2023b).

Finally, sirolimus has shown poor efficacy in AVMs. In retrospective series, sirolimus was either ineffective in extracranial AVMs, or resulted in secondary treatment resistance (Triana et al, 2017; Gabeff et al, 2019). However, several uncontrolled case series suggested that sirolimus may improve lesions size or related symptoms in patients with inoperable or refractory AVMs, either alone or as part of multimodal strategies (Chelliah et al, 2018; Maynard et al, 2019; Govindarajan et al, 2021; Durán-Romero et al, 2022). Sirolimus also showed preclinical and clinical efficacy in *PTEN*-related VMs, including AVMs (Zabeida et al, 2024; Castillo et al, 2025). A randomized controlled trial (MAV-RAPA, NCT02042326, Table 1) is underway to assess sirolimus in extracranial AVMs.

Everolimus, a derivative of sirolimus, has also been used in organ transplant and cancer therapy, then repurposed for treating tuberous sclerosis (TS) (Krueger et al, 2010; Bissler et al, 2013). A few studies reported the use of everolimus as a possibly less nephrotoxic alternative to sirolimus in VMs (Wiemer-Kruel et al, 2019; Bevacqua et al, 2019). Whether everolimus outperforms sirolimus in treating VMs remains unexplored.

Regarding safety, both sirolimus and everolimus are associated with a range of AEs related to the central role of mTOR in immune regulation and metabolic pathways. While they are less commonly linked to hyperglycemia compared to alpelisib, sirolimus and everolimus have been frequently associated with oral mucositis, dyslipidemia, leukopenia and infections (especially in association with other immunosuppressants), gastrointestinal symptoms, delayed wound healing (linked to high doses), and rash (Kaplan et al, 2014; Adams et al, 2016; Bissler et al, 2017; Gabeff et al, 2019; Freixo et al, 2020; Pithadia et al, 2020; Maruani et al, 2021; Tedesco-Silva et al, 2022; Seront et al, 2023). In the VASE trial, high-grade AEs occurred in 18% of patients (Seront et al, 2023). The impact of mTOR inhibitors alone on the incidence of infections is difficult to infer from transplantation research due to frequent combination of other immunosuppressant drugs. Though frequent in patients with TS (79%), upper respiratory tract infections associated with everolimus seem to be mostly mild (Krueger et al, 2010).

Both drugs are also associated with frequent dermatological complications. Delayed wound healing occurs in approximately 40% of patients with TS treated with sirolimus or everolimus, and 30% kidney transplant recipients (Pithadia et al, 2020; Manzia et al, 2020). Though oral mucositis was shown to occur in 3 to 29.1% kidney transplant recipients treated with everolimus (Pascual et al, 2018), an incidence ranging from 48 to 79% was reported in TS patients (Krueger et al, 2010; Bissler et al, 2013). Additionally, acneiform rash was reported in approximately 25% TS patients after 12 months on mTOR inhibitors (Bissler et al, 2017; Pithadia et al, 2020; Bissler et al, 2013), similar to the incidence found in cancer patients receiving everolimus (Ramirez-Fort et al, 2014). Interestingly, rash seems to develop during the first months of treatment (Bissler et al, 2013, 2017; Pascual et al, 2018).

Most AEs are dose-dependent, including dermatological complications (Martins et al, 2013; Kaplan et al, 2014). To improve tolerance and treatment adherence, the DESIREE trial showed that progressive dose escalation was associated with a lower incidence of mucositis in cancer patients (Schmidt et al, 2022). More recently, a randomized trial showed that low-dose sirolimus (trough 5–8 ng/mL) was as effective as high-dose regimens (10–15 ng/mL) in Kaposiform hemangioendothelioma (a rare type of vascular tumor), with fewer respiratory, dermatological, and mucosal AEs, suggesting that a lower dosage may improve tolerance without loss of efficacy in vascular anomalies (Zhou et al, 2025).

Finally, no long-term impact on growth was reported in children treated with sirolimus (Wang et al, 2024).

## TIE2-angiopoietin axis modulators

Drugs targeting the TIE2-angiopoietin pathway are being explored in retinal vascular diseases and cancer. While ANG1 inhibitors have not yet reached clinical trials (Saharinen et al, 2017; Parmar and Apte, 2021), current evidence suggests that ANG2 inhibitors may be most effective when combined with other anti-angiogenic therapies such as VEGF inhibitors (Ferro Desideri et al, 2022). Interestingly, a recent case report showed that rebastinib, a TIE2 inhibitor (Fig. 1), reduced lesion size and improve quality of life in a patient with a severe cervicofacial venous malformation (Triana and Lopez-Gutierrez, 2023). TIE2-ANG inhibitors could thus emerge as a relevant therapeutic option in VMs.

## Inhibitors of the RAS-MAPK pathway

### KRAS inhibitors

The first KRAS-selective inhibitor, sotorasib (Fig. 1), was approved for KRAS$^{G12C}$-mutated non-small cell lung carcinoma in 2021 by the US FDA (Research, 2021). In 2024, our group reported that sotorasib improved KRAS$^{G12C}$-related VMs and survival in two mouse models (Fraissenon et al, 2024). In 2 patients with inoperable AVMs, sotorasib was used compassionately at the same dosage as approved for lung cancer (Research, 2021). Both patients were carefully monitored with repeated clinical, biological (including liver enzymes) and imaging surveillance. Sotorasib led to a reduction in VM volumes by 31.5% and 19.8% after 24 and 6 months, respectively. One patient showed improvement in hearing loss resulting from previous progression of the AVM. These results reinforce the existing evidence that directly targeting causal mutations is an effective approach for oncogene-driven VMs. Indeed, the high specificity of sotorasib for KRAS$^{G12C}$ confers effective inhibition with limited off-target effects.

Tolerance was acceptable in both patients, with only grade 1 diarrhea reported in one, which resolved after dose reduction. No high-grade AEs were observed in this study, but previous oncology trials reported frequent diarrhea, nausea, abnormal hepatic tests, and fatigue (Hong et al, 2020; Skoulidis et al, 2021; Fakih et al, 2022, 2023; Strickler et al, 2023; de Langen et al, 2023). Unfortunately, because the KRAS$^{G12C}$ variant is uncommon in both cancer and VMs, most patients with AVMs will not benefit from sotorasib (Salem et al, 2022).

Additionally, a second KRAS$^{G12C}$ inhibitor, adagrasib, was also approved in cancer but has not been reported in VMs to date (Research, 2022).

### MEK inhibitors

In 2013, trametinib was the first MEK inhibitor (Fig. 1) approved by the US FDA for the treatment of BRAF-mutant melanoma (Wright and McCormack, 2013). A few years later, Nikoalev et al reported that KRAS GoF variants cause brain AVMs (Fig. 2) and increase MAPK signaling in ECs (Nikolaev et al, 2018). Fish et al demonstrated that MEK inhibition reduced MAPK signaling, reactivated angiogenic and migration pathways in KRAS$^{G12V}$-mutant HUVECs, and rescued AVMs in a KRAS-mutant zebrafish model (Fish et al, 2020). Finally, in a mouse model of

Kras$^{G12D}$-related VMs, Nguyen et al showed that trametinib improved both survival and vessel structure (Nguyen et al, 2023). In patients, several case reports supported the use of MEK inhibitors in AVMs. In a child with a large, sirolimus-refractory extracranial AVM harboring a MAP2K1 variant, trametinib reduced lesion volume (Lekwuttikarn et al, 2019). In an adult patient, a KRAS-mutant extracranial AVM decreased in flow and size after 6 months on trametinib (Edwards et al, 2020; Cooke et al, 2021). In a patient with a germline EPHB4 variant, trametinib improved high-output cardiac failure, though AVM size remained unchanged on MRI (Nicholson et al, 2022). Finally, combined trametinib and dabrafenib (a mutant-BRAF inhibitor) therapy reduced swelling in a patient with a BRAF-mutated AVM (Fraustro et al, 2023). Based on these encouraging signals, several clinical trials are underway to assess the efficacy of trametinib in brain and extracranial AVMs, either alone or as part of multimodal therapies (NCT06098872, NCT04258046, EudraCT 2019-003573-26, Table 1).

Finally, trametinib also showed effectiveness in RAS-related LMs. In vivo, it improved lymphatic valve formation in a mouse model of GSD, but not survival (Homayun-Sepehr et al, 2021). In 2 patients with KLA, trametinib improved LM volume, DIC and respiratory function within months (Foster et al, 2020; Chowers et al, 2023).

Unfortunately, the use of trametinib is limited by frequent AEs. In oncology trials, overall AEs incidence was 97% with trametinib monotherapy (including 36% grade ≥3), compared to 85% with dabrafenib monotherapy (Garutti et al, 2022). Dermatological and gastro-intestinal toxicities are the most frequent AEs of trametinib in cancer patients (Garutti et al, 2022). Indeed, trametinib was associated with acneiform (63% adults, 31% children) and maculopapular rash (42% adults, 39% children), alopecia and paronychia, diarrhea (73% adults, 46% children), and oral mucositis (35% adults, 31% children), often requiring dose adjustments in severe cases (Gershenson et al, 2022; Bouffet et al, 2023; Anforth et al, 2014). It was also associated with frequent fever (43% of adults, 31% of children) and fatigue (28% of adults, 39% of children) (Garutti et al, 2022; Bouffet et al, 2023).

Similarly, in nonmalignant conditions, a meta-analysis of trametinib in pediatric neurofibromatosis reported paronychia in 61% of cases, rash in 26%, diarrhea in 17%, and mouth ulcers in 0.5% (Wang et al, 2022). Such AEs were also reported in patients with AVMs receiving trametinib (Nicholson et al, 2022; Seebauer et al, 2024). Trametinib was also linked to decreased left ventricular ejection fraction (LVEF), peripheral edema, and severe hypertension in 6–12% of adult patients (Banks et al, 2017; Mincu et al, 2019; Glen et al, 2022; Gershenson et al, 2022; Garutti et al, 2022; Bouffet et al, 2023). In cancer treatment, trametinib combined with dabrafenib was associated with a higher incidence of venous thromboembolism than dabrafenib alone (Mincu et al, 2019). Since VM patients are at risk of cardiac failure and thromboembolism, the use of MEK inhibitors requires careful consideration and close monitoring.

## Guanine nucleotide dissociation inhibitors

The discovery of activating GNAQ and GNA11 variants in uveal melanoma (UM) and in capillary malformations has spurred the development of therapeutic strategies aimed at targeting

Gαq/11 signaling (Van Raamsdonk et al, 2010; Shirley et al, 2013). Guanine Nucleotide Dissociation Inhibitors prevent the release of GDP from Gα proteins, keeping them inactive (Fig. 1) (Bichsel and Bischoff, 2019; Silva-Rodríguez et al, 2022). Two compounds, FR900359 and YM-254890, have been shown to suppress oncogenic Gαq signaling and inhibit proliferation in diverse UM cell lines in vitro (Onken et al, 2018; Hitchman et al, 2021). To date, the use of Gαq and Gα11 inhibitors has not been reported in patients with VMs.

## Inhibitors of the VEGF pathway

The main angiogenic factor, VEGF, has sparked significant interest in oncology due to its involvement in cancer-associated angiogenesis (Carmeliet, 2005). VEGF signaling is also a relevant target in VMs. Elevated VEGF levels were reported in the serum and nasal mucosa of HHT patients (Mansur and Radovanovic, 2023). Moreover, in an *ALK1*-deficient HHT mouse model, intracranial VEGF injections accelerated AVM progression, brain hemorrhage, vascular leakage, and mortality (Cheng et al, 2019). Several inhibitors of VEGF signaling have been assessed in VMs: anti-VEGF drugs, immunomodulatory drugs (IMiDs), and multikinase RTK inhibitors (Fig. 1) (Seebauer et al, 2024). Notably, in vitro studies demonstrated that pharmacological inhibition of VEGF reduces the proliferation rates of AVM-derived ECs and inhibits the formation of cell networks (Seebauer et al, 2024).

### Anti-VEGF drugs

Bevacizumab, an anti-VEGF monoclonal antibody, has shown preclinical efficacy in HHT-related AVMs by reducing vascular EC proliferation and increasing apoptosis (Walker et al, 2012). In a 2012 study, bevacizumab decreased cardiac output and duration and frequency of epistaxis episodes in 25 HHT patients (Dupuis-Girod et al, 2012). In 3 patients with unresectable or incompletely resected sporadic AVMs—including 2 with *KRAS* somatic variants—intravenous or intralesional bevacizumab reduced bleeding, pulsatility and pain after 8 months of treatment. However, trametinib outperformed bevacizumab in patients with *KRAS* variants, improving both lesion size and symptom control (Seebauer et al, 2024). Finally, a single-arm pilot study (NCT02314377) including only two patients with large unresectable sporadic brain AVMs found stable lesion volumes after treatment with bevacizumab, despite effective VEGF blockade, and no intracranial hemorrhage or severe AEs (Muster et al, 2021). Though it may be beneficial in germline AVM syndromes, these results suggest that bevacizumab may not be the most relevant therapeutic target in sporadic AVMs. Clinical trials are currently underway to evaluate bevacizumab in brain AVM-related symptoms in patients ineligible for interventional therapy (NCT06264531, Table 1).

Additionally, aflibercept, a soluble VEGF decoy receptor, improved hemoglobin levels and reduced bleeding and transfusion needs after 6 months in an HHT patient with bevacizumab-refractory gastrointestinal bleeding (Villanueva et al, 2023). Therefore, aflibercept may offer a promising alternative for patients unresponsive or ineligible to bevacizumab.

In onco-hematology studies, bevacizumab was linked to AEs including hypertension, bleeding, thromboembolism, delayed wound healing, cardiac and cerebral ischemia (Sharma and Marcus, 2013; Totzeck et al, 2017). However, severe (grade ≥3) AEs were rare in HHT, with only 2 cases of hypertension reported among 25 patients (Dupuis-Girod et al, 2012).

### IMIDs

More recently, thalidomide was reported to reduce bleeding in germline and sporadic AVMs. In vitro studies reported that thalidomide represses VEGF synthesis and depletes VEGFRs in human ECs (Yabu et al, 2005; Komorowski et al, 2006). In HHT mouse models, thalidomide reduced bleeding and improved vascular morphology by increasing pericyte coverage and reducing VM density and inflammation (Lebrin et al, 2010; Zhu et al, 2018). Thalidomide was associated with reduced bleeding in 6 out of 7 HHT patients, and a phase 2 study confirmed these findings in 31 patients, reporting only mild drug-related AEs (Lebrin et al, 2010; Invernizzi et al, 2015). Likewise, a clinical trial recently showed that pomalidomide, a thalidomide derivative, improved epistaxis severity scores in HHT patients (Al-Samkari et al, 2024).

In 18 patients with symptomatic, refractory sporadic AVMs, thalidomide also improved pain, bleeding, skin ulcers and cardiac failure (Boon et al, 2022). However, several patients received concurrent interventions, most of them had stable disease, four experienced progression, and genetic data were lacking (Boon et al, 2022). Another study found improved symptom control in 11 patients with extracranial AVMs treated with thalidomide after embolization failure. After a median of 10 months on treatment, pain improved in 6 of 9 patients, bleeding decreased, 4 had no more visible target vessel on angiography, and 2 with high-output cardiac failure became asymptomatic (Nip et al, 2023).

Interestingly, thalidomide also showed potential in reducing bleeding in patients with small-intestinal angiodysplasia, supporting its potential relevance for treating slow-flow VMs (Chen et al, 2023a).

In patients with sporadic and germline AVMs, asthenia and erythroderma were the only reported high-grade (grade ≥3) thalidomide-related AEs (Boon et al, 2022; Invernizzi et al, 2015). Additionally, thalidomide has been linked with an increased risk of thrombo-embolism in myeloma patients, therefore requiring careful surveillance (El Accaoui et al, 2007). Finally, 47% of HHT patients treated with pomalidomide experienced high-grade AEs, including constipation, fatigue, neutropenia, and rash (Al-Samkari et al, 2024).

### Receptor-tyrosine kinase inhibitors

Other inhibitors such as pazopanib, targeting multiple RTKs including VEGFR2, are currently being assessed in severe HHT clinical trials (NCT03850730, NCT03850964, Table 1) (Mansur and Radovanovic, 2023).

Overall, anti-VEGF therapies show promising efficacy in reducing AVM-related symptoms, especially in HHT patients ineligible for other treatments. However, given the absence of obvious impact on lesion volume in sporadic AVMs, they are unlikely to be first-line options in this context.

# Challenges and future directions

## Personalized medicine for vascular malformations

Personalized medicine is becoming central to VM treatment, and new therapeutic approaches are now tailored to each patient's genetics and clinical profile. Advances in VM care are expected to arise from improvements in genetic diagnosis, multimodal and

combination therapies, and innovative approaches such as gene therapy and PROTACs.

### Improving molecular sensitivity

Advances in genetic sequencing technologies have enabled the identification of somatic variants driving VMs, paving the way for targeted therapies. However, detecting these variants remains challenging, particularly in inaccessible tissues (e.g., brain AVMs) or when variant allele frequency (VAF) is very low. Innovative techniques such as DNA sequencing on isolated lymphatic or blood ECs and cell-free DNA (cfDNA) sequencing might help address these limitations (Li et al, 2023a). Additionally, deep coverage exome sequencing and ultra-deep targeted sequencing now allow for the detection of pathogenic variants with VAFs as low as 0.15% (Li et al, 2023a; Wedemeyer et al, 2024). A key remaining question is how isolated mutations give rise to such extensive clinical manifestations. Wedemeyer et al used long-read single-cell RNA sequencing in PROS to better characterize the cell types harboring mutations and their transcriptional profiles. These findings provide insights into how small populations of mutated cells influence their microenvironment, contributing to disease development (Wedemeyer et al, 2024).

### Multimodal strategies and synergistic combinations

Complex VMs often require multimodal therapy for optimal management. While interventional treatments like surgery, sclerotherapy, or embolization can be effective, they may not be sufficient for extensive or recurrent lesions. Combining systemic targeted therapies (e.g., PI3K or mTOR inhibitors) with these approaches has shown promise in reducing lesion size, alleviating symptoms, and improving surgical outcomes for previously inoperable cases (Gits et al, 2014; Dorrity et al, 2022; Schmidt et al, 2023).

Recent case studies highlight the potential of combination therapies, such as sirolimus and trametinib, in treating severe CLAs or AVMs (Liu et al, 2022a; Seront et al, 2024). However, combination regimens carry a higher risk of adverse effects. Oncology studies assessing dual MAPK and PI3Kα inhibition have frequently reported dose-limiting toxicities, underscoring the need for careful dosing and monitoring (McNeill et al, 2017; Schram et al, 2018; Ramanathan et al, 2020).

### Addressing treatment resistance

To date, no case of acquired resistance has been reported in patients treated with targeted therapies aimed at causal mutations (e.g., alpelisib in *PIK3CA*-related VMs or sotorasib in KRAS[G12C]-driven VMs) (Venot et al, 2018; Delestre et al, 2021; Pagliazzi et al, 2021; Garneau et al, 2021; Kolitz et al, 2022; Morin et al, 2022; Garreta Fontelles et al, 2022; Sterba et al, 2023; Canaud et al, 2023; Zerbib et al, 2024; Fraissenon et al, 2024; Cossio et al, 2024; Etingin et al, 2025). Resistance mechanisms probably differ from what is seen in cancer, due to the usual absence of concurrent mutations. Indeed, long-lasting responses are observed with inhibitors like alpelisib in PROS or sotorasib in KRAS[G12C]-related VMs, in contrast to cancer (Venot et al, 2018; Tanaka et al, 2021; Canaud et al, 2023; Fraissenon et al, 2024). Unfortunately, the effects of pharmacological inhibitors are typically transient: indeed, we observed disease progression following alpelisib discontinuation in three pregnant PROS patients. However, no evidence of secondary resistance was observed upon treatment reinitiation after delivery (Morin et al, 2025a).

### Drug delivery systems

Systemic therapy faces several limitations, including low bioavailability, short half-life, insufficient drug accumulation at target sites, off-target distribution, and toxicity to healthy tissues. Consequently, pharmacokinetic optimization may enhance target engagement and decrease off-target effects. Transdermal drug delivery systems, which avoid oral administration, may reduce systemic exposure and enhance local efficacy (Geng et al, 2024). Indeed, topical sirolimus improved superficial VMs in pediatric patients, albeit with limited efficacy due to poor transdermal drug penetration (Dodds et al, 2020). To overcome this challenge, nanoparticles and microneedle patches may offer pharmacokinetic solutions by improving dermal delivery (Waghule et al, 2019; Zeng et al, 2020). Finally, nanodrug delivery systems that target nanoparticles to specific cellular markers—such as VEGFR— to increase drug concentration in target cells may enhance treatment specificity. For instance, exosome-mimetic nanoparticles carrying rapamycin have been shown to extend its half-life in the human vascular network (Li et al, 2022). Alternatively, photothermal therapy after infusion of CD31-targeted gold nanorods has shown promising results in venous malformations in vivo (Jiang et al, 2022). This approach enables targeted drug delivery to specific cells with precise spatial control using near-infrared light.

## Emerging therapies

### Improving tolerance: mutant-selective PI3Kα inhibitors

PI3Kα inhibitors are linked to several AEs, including glucose metabolism disorders resulting from wild-type PI3Kα inhibition. To overcome this limitation, mutant-selective PI3Kα inhibitors are being developed. In patient-derived xenograft models, STX-478, a PI3Kα[H1047R]-selective inhibitor, reduced tumor volume with minimal impact on glucose metabolism (Buckbinder et al, 2023). Likewise, RLY-2608, a pan-mutant-selective PI3Kα inhibitor, is under clinical evaluation (NCT05216432) (Saura et al, 2024).

### Drugging the undruggable: mutant-selective and pan-(K)RAS inhibitors

RAS was long considered undruggable due to the small size of docking sites at its surface, making it difficult to design specific inhibitors (Kessler et al, 2019). A change of paradigm occurred with the approval of the first mutant-selective RAS inhibitors, sotorasib and adagrasib. Many drugs are currently under clinical evaluation, including several KRAS[G12D]-selective inhibitors (Cox and Der, 2024). However, while mutant-selectivity was expected to enhance both efficacy and tolerance, high-grade AEs still occurred in 35% of patients treated with sotorasib, including diarrhea, fatigue, nausea or hepatotoxicity (Hong et al, 2020; Fakih et al, 2022, 2023; Strickler et al, 2023; de Langen et al, 2023).

Additionally, the first pan-(K)RAS inhibitors are currently assessed in oncology clinical trials (Kim et al, 2023; Jiang et al, 2024; Holderfield et al, 2024; Cox and Der, 2024). The pan-KRAS inhibitor BI-2865 targets most KRAS variants (as well as the wild-type protein) (Kim et al, 2023). As an off-state inhibitor, it targets GDP-bound KRAS. The analog BI-3706674 is currently assessed in KRAS[WT]-amplified and KRAS-mutant cancer (NCT06056024).

Finally, pan-RAS inhibitors offer the broadest activity spectrum. The RMC-6236 on-state inhibitor targets all three RAS paralogs, including mutant and wild-type proteins, and forms an inhibitory tricomplex with cyclophilin A, an intracellular chaperone. It is currently evaluated in a phase 1 study (NCT05379985), showing promising preliminary results in pancreatic and lung cancer (Arbour et al, 2023; Jiang et al, 2024).

Particular attention will focus on the tolerance profiles of these new RAS inhibitors, as they lack variant and isoform specificity. If efficacy and safety are acceptable, they could revolutionize the treatment of RAS-related VMs.

### Breaking RAS-PI3Kα interaction

For several reasons, disrupting RAS-PI3Kα interaction is an interesting strategy in VMs. First, RAS-PI3Kα interaction is required for RAS-driven tumorigenesis, tumor-induced angiogenesis, and metastasis (Gupta et al, 2007; Castellano et al, 2013; Murillo et al, 2014). Moreover, RAS is required for the activation of the E542K and E545K PI3Kα variants (Czyzyk et al, 2025). Finally, because insulin signals independently of RAS-PI3Kα interaction, drugs that break this interaction are unlikely to induce glucose metabolism disorders (Taha and Klip, 1999; Liu et al, 2006; Simanshu et al, 2025). The BBO-10203 drug blocks the interaction between RAS and the PI3Kα RBD, inhibiting PI3Kα signaling in both wild-type and *PIK3CA*-mutant cancer cell lines. It effectively reduced the growth of human tumor xenografts in vivo, including those harboring *PIK3CA* GoF variants, without impact on blood glucose levels (Beltran et al, 2024; Simanshu et al, 2025). It is currently assessed in a phase 1 clinical trial for advanced breast, colorectal, and lung cancer (NCT06625775, BREAKER-101). Depending on its tolerance profile, future studies might address its efficacy in *RAS*- or *PIK3CA*-mutant VMs.

### Degrading mutant proteins with PROTACs

Proteolysis-targeting chimeras (PROTACs) drive proteasome-mediated degradation of target proteins. A key advantage of these inhibitors is their recycling after target protein degradation, enabling durable inhibition. However, their high molecular weight may require intravenous administration (Békés et al, 2022). Several compounds aimed at major proteins involved in VMs are currently under development. WJ112-14 is a specific PI3Kα degrader, with high selectivity towards the α isoform, and very low affinity towards PI3Kβ. Indeed, it was suggested that alpelisib-related metabolic AEs may result from nonspecific inhibition of PI3Kβ; increasing isoform selectivity may therefore improve metabolic tolerance (Jauslin et al, 2024). Another PI3Kα-targeted PROTAC, ZM-PI05, degrades both PI3Kα subunits (p110α and p85) and shows a lower IC50 than alpelisib in inhibiting proliferation of *PIK3CA*-mutant breast cancer cell lines (Zhang et al, 2024). RAS-targeted PROTACs are also underway, with ACBI3 degrading most of the common KRAS mutants and inducing tumor regression in vivo (Popow et al, 2024).

### Gene editing

Rather than inhibiting mutant proteins, new gene therapy approaches focus on innovative methods to edit somatic variants. Attempts of gene editing in *KRAS* have been reported in cancer studies using Clustered Regularly Interspaced Short Palindromic Repeats (CRISPR)/CRISPR associated protein 9 (Cas9) technology to generate double-strand breaks and disrupt variant *KRAS* alleles

(Lee et al, 2018; Kim et al, 2018). Despite initial tumor cell depletion, the emergence of treatment-insensitive *KRAS*-mutant clones motivated the refinement of the technique by using base editing in cancer. Base editing seems more effective by correcting the most frequent single-nucleotide variants in the *KRAS* gene in vitro, with incomplete but promising efficacy (Sayed et al, 2022; Jang et al, 2023). Unfortunately, base editing is currently limited by the uncontrolled risk of off-target effects. The occurrence of bystander edits in *KRAS* wild-type cells may indeed result in unwanted effects. Gene editing is currently being evaluated in familial hypercholesterolemia (VERVE-101, NCT05398029), paving the way for innovative therapies targeting cancer—and potentially VMs—to enter clinical trials in the coming years (Han, 2024).

## Conclusion

Given the clinical and genetic heterogeneity of VMs, defining standardized treatment strategies remains challenging. However, advances in genetic sequencing and drug development have opened new avenues for precision medicine. Targeted therapies that directly inhibit mutant proteins offer a promising alternative to traditional symptomatic treatments. Multiple studies have already highlighted the potential of these strategies in specific VM subtypes, with ongoing clinical trials strengthening the evidence. Insufficient efficacy or resistance, especially when targeting downstream effectors, may be overcome by synergistic combination therapies, though often at the expense of tolerance. As new oncology drugs continue to enter the clinical landscape, the therapeutic arsenal for VMs is likely to evolve swiftly. Therefore, clinicians and researchers invested in the field should stay informed of therapeutic advances to better manage these complex disorders.

### Pending issues

- Identifying modifier genes involved in second-hits in germline disorders.
- Exploring non-cell autonomous mechanisms, intercellular communication and behavioral changes caused by mosaicism in postzygotic diseases.
- Assessing the long-term AEs of oncology drugs used in VMs.
- Evaluating the efficacy of various inhibitors compared with a reference drug (e.g., sirolimus) rather than placebo.
- Assessing innovative oncology therapies: RAS inhibitors, PROTACs, gene therapies are currently (or soon to be) evaluated in clinical trials.

## Peer review information

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

## Acknowledgements

This study was supported by the European Research Council (CoG 2020 grant number 101000948 awarded to GC), the Agence Nationale de la Recherche – Programme d'Investissements d'Avenir (ANR-18-RHUS-005 to GC), the Agence Nationale de la Recherche – Programme de Recherche Collaborative (19-CE14-0030-01 to GC), the Agence Nationale de la Recherche – Chaire d'Excellence France 2030 (ANR-25-CHBS-0008 to GC), and Fondation pour la Recherche Médicale (FDM202006011222) awarded to GM. We are also very grateful to our generous donors. The authors sincerely thank Dr Antoine Fraissenon (Service de Radiologie, Hospices Civils de Lyon, Lyon, France) for providing the imaging data shown in Fig. 2.

## Author contributions

**Gabriel Morin**: Conceptualization; Data curation; Methodology; Writing—original draft; Writing—review and editing. **Ilaria Galasso**: Conceptualization; Data curation; Validation; Visualization; Writing—original draft; Writing—

review and editing. **Guillaume Canaud**: Conceptualization; Data curation; Supervision; Funding acquisition; Validation; Investigation; Methodology; Writing—review and editing.

## Disclosure and competing interests statement

A patent application ("BYL719 (alpelisib) for use in the treatment of PIK3CA-related overgrowth spectrum" #WO2017140828A1) has been filed by INSERM (Institut National de la Santé et de la Recherche Médicale), Centre National De La Recherche Scientifique (CNRS), Université Paris Cité, and Assistance Publique-Hôpitaux De Paris (AP-HP) for the use of BYL719 (alpelisib) in the treatment of *PIK3CA*-related overgrowth spectrum (PROS/CLOVES syndrome). GC is the inventor. This patent is licensed to Novartis. GC receives or has received consulting fees from Novartis, Fresenius Medical Care, Vaderis, Alkermes, IPSEN and BridgeBio. The other authors declare no other competing interests. GC is an editorial advisory board member.

