## [Peer Review File · EMBO Molecular Medicine]

To Vascular Malformations: From Genetics to Therapeutics

Guillaume Canaud, Gabriel Morin, and Ilaria Galasso

Corresponding author: Guillaume Canaud (guillaume.canaud@inserm.fr)

Review Timeline:

Submission Date:	12th Mar 25
Editorial Decision:	11th Apr 25
Revision Received:	17th Sep 25
Editorial Decision:	23rd Oct 25
Revision Received:	5th Nov 25
Accepted:	10th Nov 25

Editor: Lise Roth

Transaction Report:

11th Apr 2025

Dear Guillaume,

Thank you for submitting your review to EMBO Molecular Medicine. We have now received feedback from the experts who have agreed to evaluate your manuscript.

As you will see from their reports below, they found the review interesting and well written overall. However, they do raise some concerns and make some suggestions to improve the interest and impact of your work.

We would therefore welcome a revised version of your manuscript that addresses the reviewers' points. As mentioned by reviewer #2, please ensure that the review is balanced with regard to alpelisib and sirolimus, and refrain from overstatements related to potential conflicts of interest. Please include a cover letter detailing how you have addressed each of the points raised by the reviewers.

- 1/ A .doc formatted version of the manuscript text (including Figure legends and tables)
- 2/ Separate figure files
- 3/ A letter INCLUDING the reviewer's reports and your detailed responses to their comments.
- 4/ A glossary: EMBO Molecular Medicine articles are accompanied by a glossary explaining some of the terms used for laymen.
- 5/ Pending issues: At the end of each article, there is a box highlighting issues that still need further studies and where research efforts should converge (called the Pending issues box).
- 6/ A 'disclosure statement and competing interests' statement (<https://www.embopress.org/competing-interests>).

For the figures:

We work with one of our expert scientific illustrators, who will assist with getting the figures to a publication ready state.

Please also note the following points:

- If there are certain aspects of your figure draft that are based upon assumptions or where the scientific data remains ambiguous, please add a comment so that we can work with you on an accurate depiction. Please ensure the directionality and nature of interactions is presented accurately.
- If the figure or single panels of the figure have been adapted from a published figure, please add this information to the figure legend (e.g., 'Adapted from...' or 'Based on...').
- Please only re-use figures or parts of a figure if this is essential for understanding the concept communicated. Often a reference to a previous paper will suffice. If the figure contains re-used images or elements of images, please make sure that you have the permission/license to publish it (this also applies to your own previous work, if the journal you published in retains copyright. Certain 'creative commons' open access licenses, such as CC-BY 4.0, allow re-use without additional formal permissions). All re-used material must be explicitly cited.
- If you use an image data base for scientific iconography (e.g., BioRender), please let us know if you have a license that allows for publication in an academic journal. Often authors use misleading iconography for expedience. Please ensure the information shown is scientifically accurate.
- For figures created using a software for editing vector objects like Inkscape, CorelDraw etc., please send the file as a PDF (or SVG, or EPS), PowerPoint or Keynote in which the labels and objects are still editable. For figures created using Adobe Illustrator, please send the Illustrator (.ai) file.

Looking forward to receiving your revised manuscript,

With kind regards,

Lise

**** Reviewer's comments ****

Referee #1 (Remarks for Author):

This is a very educational and detailed/updated review for VMs: molecular pathology and recent therapeutic trials. I have only a few minor requests to the authors.

1. in figure 1, it will be better for readers to understand if the authors includes CCMs and their link to RAS-MAPK pathway. In addition, it will be great if each pathway (AKT-mTOR, RAS-MAPK, TGF-b etc) is marked (or noted) in the figure.
2. in Therapeutics, p10-11, it will be great if the authors can mention about alpelisib efficacy in MCAP.

Referee #2 (Remarks for Author):

In this manuscript, the authors present a comprehensive review of the most recent theragnostic strategies in vascular malformations. While this work is largely well written and covers an acceptable and balanced level of overview and detail, I have the following major concerns that need to be addressed:

1. My major concern is that this work is clearly biased towards the last author's conflict of interest (COI) regarding alpelisib. This is exemplified by the fact that alpelisib appears right upfront in the abstract as a treatment agent for slow-flow malformations while more widely used rapalogs are completely left out.

Even more concerning is the fact that (S)AEs of alpelisib are clearly understated while those of sirolimus are overstated according to recently published work as well as unpublished real-world data on alpelisib from different institutions. These data demonstrated that alpelisib that (S)AE occur very frequently (and not necessarily dose dependent) and include severe eczema (not just a rash), and hyperglycemia, and, most importantly, an impaired increase in weight and height in prepubertal patients. At our institution, pediatric patients on alpelisib are significantly impacted regarding their growth curves and 50% of the adult patients are on metformin. This needs to be clarified and good surveillance under alpelisib stressed more. Further, it would be beneficial to add more citations in lines 322-327 other than the author's previous work (except for Andre et al). Please work on the (S)AE part for sirolimus. While it indeed has side effects, the impaired glucose metabolism resulting in diabetes is super rare (we never had a patient that developed DM2) and with good trough level controls, sirolimus is not that difficult to manage.

2. Along these lines, the author's description of sotorasib requires a more critical approach. What was the surveillance like? In oncology liver enzymes are frequently elevated under sotorasib and it is hard to believe that there were no changes whatsoever in VM patients. What percentage of the usual oncology dose was used in these patients?

Minor comments:

- It is confusing that vascular anomalies and vascular malformations are mentioned interchangeably. If this should not include vascular tumors, it would be good to be more precise regarding the terminology in the introduction.
- Line 69: Suggest genetic variants instead of anomalies
- Line 85: Please cite the original publication
- Line 98: Suggest malignant tumor entities instead of cancer (the latter is diffuse from an oncology standpoint)
- Lines 110-111: vasculogenesis followed by angiogenesis = vascular development > this part can be improved and more clear from a vascular biology standpoint
- Lines 130-133: This paragraph needs work: mention autocrine and paracrine Ang/Tie signaling, please add citations
- Paragraph C. a. could be shortened and structured in a more stringent manner
- Lines 225 ff: DVIC should be replaced by DIC - the official term used in hematology. In terms of coagulopathies LIC needs to be mentioned as well and, specifically for KLA, the Kasabach-Merritt phenomenon is missing and should be added (as it precedes DIC in KLA).
- Line 239 ff: Please use high-output cardiac failure throughout
- Lines 434 ff: It would be good to mention the relatively high organ toxicity of trametinib compared to dabrafenib
- Lines 467 ff: Define bevacizumab first
- In the conclusion, the importance of combinational targeted strategies could be emphasized
- Figure 1: Please add a nucleus and I would suggest using the terms differentiation and proliferation instead; please add legend of signs used
- Figure 2C: The cMRI of the AVM could be more representative. For AVMs, it is generally most representative to show a TWIST Angio to appreciate the full extent of the lesion
- Genetics Table in Figure 2: This is not totally clear to me: e.g., AVMs can be isolated and syndromic and harbor far more mutations including HRAS, NRAS, ARAF and even PIK3CA

Referee #3 (Remarks for Author):

The authors review the current knowledge of vascular malformations. Overall, this is a timely and interesting topic of research. While the authors have done a good job summarizing relevant information, some aspects require attention before publication. Particularly, there are many inconsistencies regarding references; authors should carefully revise those.

Abstract: In line 27, include rapamycin, as extensive work on this drug and VMs has been done and published.

In line 54, the authors include the incidence of VM and AVM but not of LM

Line 79, please put a reference at the end of the sentence (...additional events)

At the end of 79-82 sentences, add this publication Limaye N Nature Genetics 2009 (Vikkula lab)

In line 91, what do the authors mean with VM syndromes? And why KRAS-related AVMs are considered so?

Please rewrite the sentence in line 117, it is not clear.

Line 125, please consistently use the term PROS in the review, instead of other clinical syndromes (KTS). If the specific word of KTS wants to be referred to, explain is a subtype of PROS.

Line 130-131, include references at the end of the sentence

In line 141, please change the word recruit to activate. PIP3 does not recruit mTORC2...

In line 152, Makinen et al. is not the correct reference to the above statement.

Please change it. Also, I am not sure if the link between ANG-2 and PDGF with disruption of VSMCs has been shown. If so, include a reference

At the end of line 150, add the preprint Sabata et al.
(doi: <https://doi.org/10.1101/2025.02.25.640041>)

In line 157, the reference Luks et al is incorrect. The authors should replace it with Castillo et al. 2016, Castel et al. 2016, and Limaye et al. 2015.

In line 158, the reference Kobialka et al is incorrect. Please cite a LM paper here

In line 163, Luks et al is wrong here, add instead Castillo et al. 2025(<https://aacrjournals.org/cancerdiscovery/article-abstract/doi/10.1158/2159-8290.CD-24-0807/754460/Somatic-uniparental-disomy-of-PTEN-in-endothelial?redirectedFrom=fulltext>)

Add a reference which refers to the angiogenic secretome in line 166

In line 168, the reference Graupera et al incorrect (this is a LOF model). Please change it to Hare et al. 2015 (from the Wayne Phillips lab).

Please re-write the sentence in line 177; "The CCM complex....). While the authors summarize the scheme of that paper (Ren et al.)... Currently, it is not fully elucidated how the different signalling components interact. The way that the sentence is written implies this is indeed what happens, and at this stage, it is just a proposition based on the data of Ren et al. Please turn down the tone at least

The sentence in 183 is incorrect. There is a second hit in some cases, but not in all. Please correct

Something is missing at the end of the sentence starting in line 206...

For the NRAS Q61R, cite Manevitz-Mendelson et al. (Angiogenesis 2018), as this is the first manuscript in which the mutation is identified in KLA.

In line 263, please use activate instead of recruit. Also, PLCb activates RAS-MAPK but not PI3K.

In line 269, R183Q does not result in constitutive activation of GNAQ; what it does is sustain the GTP loading, which sustains the activation longer. Correct.

In Shirley et al., the authors show that pAkt is not altered upon expressing R183Q in ECs. Thus, the sentence in line 269 is incorrect: GNAQ does not dysregulate the PI3K pathways, similarly to PIK3CA mutations.

In line 273, the GNAQ Q209L variant is not the only causing congenital hemangioma.

In line 296, what does it mean suspensive?

In line 303, mutations in PIK3R1 have been published (<https://www.sciencedirect.com/science/article/pii/S1769721222001719?via%3Dihub>)

I would suggest using PIK3CA instead of p110a in line 302. This would help clinicians to read through. Also, be consistent with p110a or PI3Ka (both are fine as long as they are defined) but I would suggest not mixing (e.g, line 302, 309)

The authors should be clearer about the side effects and growth effects of Alpelisib. It is now well known that some patients, including PROS patients, cannot tolerate this treatment for a long time and that the growth effects are an issue.

In line 328, change Mirasentib for Akt inhibitors and include capivasertib since the last publication by Castillo et al. shows efficacy in PHTS-related vascular, malformations in preclinical models.

In line 333, the authors refer to preclinical studies, but the reference pertains to clinical studies.

Given that Everolimus is the best-studied drug in these conditions, the review would benefit from including some discussions about on- and off-treatment, the impact of different doses, and whether the AEs are dose-dependent.

In line 365, include this information: Sirolimus has proven preclinical and clinical therapeutic efficacy in PHTS-related vascular malformations (<https://pubmed.ncbi.nlm.nih.gov/39166269/> and Castillo et al. 2025 <https://aacrjournals.org/cancerdiscovery/article-abstract/doi/10.1158/2159-8290.CD-24-0807/754460/Somatic-uniparental-disomy-of-PTEN-in-endothelial?redirectedFrom=fulltext>)

In line 434, the authors state that trametinib shows frequent side effects. How frequent is that compared to Alpelisib? Can authors also provide information on the frequency of Alpelisib?

In line 459, I do not understand what the authors mean by the sentence! Though VEGF signals.... Please explain better. The whole section on VEGF needs some attention.

What do the authors mean by "indirect inhibitor of VEGF" in line 490?

In line 515, include the use the notion that Aflibercept has also been tested for HHT (<https://www.mdpi.com/1648-9144/59/9/1533>)

Referees' comments

Referee #1 (Remarks for Author):

This is a very educational and detailed/updated review for VMs: molecular pathology and recent therapeutic trials.

We sincerely thank the Referee for their thoughtful comment.

I have only a few minor requests to the authors.

1. in figure 1, it will be better for readers to understand if the authors includes CCMs and their link to RAS-MAPK pathway. In addition, it will be great if each pathway (AKT-mTOR, RAS-MAPK, TGF- β etc) is marked (or noted) in the figure.

We thank the Referee for this constructive comment. For clarity, we have removed the terms "RASopathies" and "PIKopathies" from the figure, as several disorders – including CCMs – may involve both pathways. As suggested, we have also highlighted the main pathways involved in vascular malformations.

2. in Therapeutics, p10-11, it will be great if the authors can mention about alpelisib efficacy in MCAP.

As requested, we have added a paragraph detailing the available evidence on alpelisib efficacy in MCAP and have mentioned the ongoing SESAM study (Luu et al. 2024a):

Page 11: "Finally, the SESAM study currently assesses alpelisib in megalencephaly-capillary malformation-polymicrogyria syndrome (MCAP, a PROS subtype), following reports of neurological and MRI improvements in treated patients (Venot et al. 2018a; Morin et al. 2022; Luu et al. 2024b)."

Referee #2 (Remarks for Author):

In this manuscript, the authors present a comprehensive review of the most recent theragnostic strategies in vascular malformations. While this work is largely well written and covers an acceptable and balanced level of overview and detail, I have the following major concerns that need to be addressed:

1. My major concern is that this work is clearly biased towards the last author's conflict of interest (COI) regarding alpelisib. This is exemplified by the fact that alpelisib appears right upfront in the abstract as a treatment agent for slow-flow malformations while more widely used rapalogs are completely left out.

We thank the Referee for their thoughtful comments, and we acknowledge the Referee's concern about the early mention of alpelisib in the abstract. Our intent was not to overemphasize this drug, but rather to reflect recent progress in the field – especially the emergence of alpelisib as the first FDA-approved drug for vascular malformations. That said, we understand the importance of balanced representation and have revised the abstract accordingly to highlight the role of other inhibitors, including sirolimus which remains widely used in clinical practice.

We fully agree that maintaining a balanced and objective perspective on all available therapeutic options is essential, particularly in the context of a review article. We have therefore revised the article thoroughly and carefully detailed both benefits and drawbacks of every treatment discussed in the text.

Even more concerning is the fact that (S)AEs of alpelisib are clearly understated while those of sirolimus are overstated according to recently published work as well as unpublished real-world data on alpelisib from different institutions. These data demonstrated that alpelisib that (S)AE occur very frequently (and not necessarily dose dependent) and include severe eczema (not just a rash), and hyperglycemia, and, most importantly, an impaired increase in weight and height in prepubertal patients. At our institution, pediatric patients on alpelisib are significantly impacted regarding their growth curves and 50% of the adult patients are on metformin. This needs to be clarified and good surveillance under alpelisib stressed more. Further, it would be beneficial to add more citations in lines 322-327 other than the author's previous work (except for Andre et al).

We thank the Referee for their remarks regarding the tolerance profile of alpelisib. As suggested, we have detailed more extensively the adverse events linked to alpelisib.

Skin toxicity, as reported by the European Medicines Agency, is very frequently associated with alpelisib. Skin rash is the most reported dermatological AE (maculopapular or

generalized). Severe toxidermia was also reported, albeit with a lower incidence. In a cohort of 102 breast cancer patients treated with alpelisib, dermatological complications were observed in 40%, and grade 3 rash in 20% (D. G. Wang et al. 2020). Only one suspicion of DRESS was confirmed, and no Stevens-Johnson/Toxic Epidermal Necrolysis syndromes were reported. Finally, 91% of patients with low-grade rash could continue alpelisib and 56% of those who interrupted alpelisib because of grade 3 events could resume treatment at the same dose without recurrence of toxidermia.

The concerns raised by the referee in pediatric populations are indeed relevant. However, based on our institutional experience using low-dose alpelisib, as well as that of others (Sterba et al. 2023a), we have observed a limited toxicity profile. So far, we have identified in our cohort growth delay in 3 children who additionally exhibited low GH levels due to megalencephaly with hypophysial involvement at baseline (*report in preparation*). Notably, growth patterns were normal in both infants we treated with alpelisib (receiving 2.8 and 3.1mg/kg/d) (Morin et al. 2022), and no warning has been issued by health authorities to our knowledge. However, we acknowledge that the scientific community has recently raised concerns regarding the long-term impact of alpelisib on pediatric growth. We took into account the report of (Etingin et al. 2025), who observed growth restriction in a 2 year-old infant who exhibited the highest drug exposure (3.6mg/kg/d) in a cohort of 8 patients (Etingin et al. 2025). Similarly, (Cossio et al. 2024) reported growth delay (-2DS) in an infant treated with 25mg/d alpelisib (4.8mg/kg/d). Overall, these results suggest that growth delay may result from alpelisib overdose (with an approximate threshold around 3mg/kg/d). We agree with the recent report by (Remy et al. 2025), that the optimal weight-adjusted alpelisib dosage should be determined in the pediatric population using pharmacokinetics data. We have discussed more extensively the emerging questions on alpelisib tolerance and referenced as exhaustively as possible the existing evidence. Nevertheless, while we reviewed the available literature, we have only found a few reports of growth delay, suggesting that it is either very rare or underreported.

In adult patients, we found alpelisib to be well tolerated, though a few patients indeed developed glucose metabolism disorders that may require the use of metformin. However, in our experience, such cases are very rare and typically occur in the presence of additional risk factors for diabetes (e.g. obesity, family history, or concomitant medications). It is therefore particularly informative to learn that the Referee's experience differs markedly from ours. We hypothesize that dietary, geographical, or ethnic factors may contribute to the observed discrepancies, in addition to the on-target adverse effects of alpelisib on insulin signaling.

Please work on the (S)AE part for sirolimus. While it indeed has side effects, the impaired glucose metabolism resulting in diabetes is super rare (we never had a patient that developed DM2) and with good trough level controls, sirolimus is not that difficult to manage.

We acknowledge that some of the sirolimus-related adverse events were overstated in the previous version of that section. We have carefully revised it as suggested by the Referee. As for alpelisib, we have reviewed articles related to both vascular malformations, organ transplant and cancer, the main diseases mTOR inhibitors have been approved for.

The risk of diabetes was particularly underscored because we cited supporting evidence coming from the transplantation field – where additional risk factors for diabetes are at play (Johnston et al. 2008). However, apart from kidney transplant, sirolimus is also associated with diabetes in cancer patients. Even though theoretical, we chose to address that risk but explicitly mentioned that such complications are less frequent than with alpelisib. Finally, we have expanded the section dedicated to dermatological and infectious complications.

“Regarding safety, both sirolimus and everolimus are associated with a range of adverse events related to the central role of mTOR in immune regulation and metabolic pathways. While they are less commonly linked to hyperglycemia compared to alpelisib, sirolimus and everolimus have been frequently associated with oral mucositis, dyslipidemia, leukopenia (especially in association with other immunosuppressants), gastrointestinal symptoms, infections, delayed wound healing and rash (Adams et al. 2016; Bissler et al. 2017; Gabeff et al. 2019; Freixo et al. 2020; Pithadia et al. 2020; Maruani et al. 2021; Tedesco-Silva et al. 2022; Seront et al. 2023). In the VASE trial, high-grade AEs occurred in 18% of patients (Seront et al. 2023). The impact of mTOR inhibitors alone on the incidence of infections is difficult to estimate due to the frequent combination of other immunosuppressant drugs in transplantation. Though frequent in patients with TS (79%), upper respiratory tract infections associated with everolimus seem to be mostly mild (Krueger et al. 2010).

Both drugs are also associated with frequent dermatological complications. Delayed wound healing occurs in approximately 40% of patients with TS treated with sirolimus or everolimus, and 30% kidney transplant recipients (Pithadia et al. 2020; Manzia et al. 2020). Though oral mucositis was shown to occur in 3 to 29.1% kidney transplant recipients treated with everolimus (Pascual et al. 2018), a higher incidence ranging from 48% to 79% was reported in TS patients (Krueger et al. 2010; Bissler et al. 2013). Additionally, acneiform rash was reported in approximately 25% TS patients after 12 months on mTOR inhibitors (Bissler et al. 2017; Pithadia et al. 2020; Bissler et al. 2013), similar to the incidence found in patients with cancer receiving everolimus (Ramirez-Fort et al. 2014). Interestingly, it seems to develop during the first months of treatment (Pascual et al. 2018; Bissler et al. 2013; 2017). Most AEs are dose-dependent, including dermatological complications (Kaplan et al. 2014; F. Martins et al. 2013). To improve tolerance and treatment adherence, the DESIREE trial showed that progressive dose escalation was associated with a lower incidence of mucositis in cancer patients (Schmidt et al. 2022).

Finally, no long-term impact on growth was reported in children treated with sirolimus (Y.-Y. Wang et al. 2024).”

2. Along these lines, the author's description of sotorasib requires a more critical approach. What was the surveillance like? In oncology liver enzymes are frequently elevated under sotorasib and it is hard to believe that there were no changes whatsoever in VM patients. What percentage of the usual oncology dose was used in these patients?

In the section addressing sotorasib in KRAS^{G12C}-related vascular malformations, we summarized the results of our study published in (Fraissenon et al. 2024).

The initial dose used was 100% of the approved dose for lung cancer (960mg/d). All adverse events were reported in the original publication. Liver enzymes were carefully monitored and remained in the normal range. We would like to remind the Referee that the low number of patients we treated (only 2) may explain why we did not observe any anomalies in hepatic tests. Indeed, we expect that with increasing numbers of patients treated, we may observe similar adverse events as observed in oncology studies. We are convinced that only selected patients should receive sotorasib after multidisciplinary discussions, and that monitoring should be very careful.

We would like to emphasize that we had explicitly mentioned that “No high-grade AEs were observed in this study, but previous oncology trials reported frequent diarrhea, nausea, abnormal hepatic tests, and fatigue (Skoulidis et al. 2021; Hong et al. 2020; Fakih et al. 2022; 2023; Strickler et al. 2023; de Langen et al. 2023)”.

However, dose-dependent grade 1 diarrhea was reported in the 2nd patient (which resolved after dose reduction) and we acknowledge that this information was missing in the previous version of the manuscript.

The sotorasib section now appears as follows:

“The first KRAS-selective inhibitor, sotorasib (**Figure 1**), was approved for KRAS^{G12C}-mutated non-small cell lung carcinoma in 2021 by the US FDA (Research 2021). In 2024, our group reported that sotorasib improved KRAS^{G12C}-related VMs and survival in two mouse models (Fraissenon et al. 2024). In 2 patients with inoperable AVMs, sotorasib was used compassionately at the same dosage as approved for lung cancer (Research 2021). Both patients were carefully monitored with repeated clinical, biological (including hepatic liver enzymes) and imaging surveillance. Sotorasib led to a reduction in VM volumes by 31.5% and 19.8% after 24 and 6 months, respectively. One patient showed improvement in hearing loss resulting from previous progression of the AVM. These results reinforce the existing evidence that directly targeting causal mutations is an effective approach for oncogene-driven VMs. Indeed, the high specificity of sotorasib for KRAS^{G12C} confers effective inhibition with limited off-target effects. Tolerance was acceptable in both patients, with only grade 1 diarrhea reported in one, which resolved after dose reduction. No high-grade AEs were observed in this study, but previous oncology trials reported frequent diarrhea, nausea, abnormal hepatic tests, and fatigue (Skoulidis et al. 2021; Hong et al. 2020; Fakih et al. 2022; 2023; Strickler et al. 2023; de Langen et al. 2023). Unfortunately, because the *KRAS*^{G12C} variant is uncommon in both cancer and VMs, most patients with AVMs will not benefit from sotorasib (Salem et al. 2022). A second KRAS^{G12C} inhibitor, adagrasib, was also approved in cancer but was not reported in VMs to date (Research 2022).”

Minor comments:

- It is confusing that vascular anomalies and vascular malformations are mentioned interchangeably. If this should not include vascular tumors, it would be good to be more precise regarding the terminology in the introduction.

Throughout the manuscript, we used the terminology published by the international society for the study of vascular anomalies (ISSVA) in 2025 (“Classification | International Society for the Study of Vascular Anomalies,” n.d.)

Therefore, wherever the term “anomaly” has been used, it refers to pathologies that fall in the spectrum of vascular anomalies, either tumors or malformations. Given that most early publications on therapeutics reported both patients with vascular tumors and malformations, and that misclassification has been frequent, it is sometimes difficult to use more specific terms than vascular anomalies. When we could assert that the referenced publications had carefully selected patients with vascular malformations (and not tumors), the term “malformation” was used.

- Line 69: Suggest genetic variants instead of anomalies

We thank the Referee for mentioning this mistake. We have revised the text accordingly.

- Line 85: Please cite the original publication

We have replaced the previously cited references with a 1978 publication on segmental skin disorders that includes an abstract indexed in Medline (Happle 1978). Although among the first reports on the topic, the earlier references lacked abstracts in PubMed.

- Line 98: Suggest malignant tumor entities instead of cancer (the latter is diffuse from an oncology standpoint)

We agree with the Referee. We have changed the text accordingly.

- Lines 110-111: vasculogenesis followed by angiogenesis = vascular development > this part can be improved and more clear from a vascular biology standpoint

We thank the Referee for this relevant comment. We have modified the text accordingly for clarity (“angiogenesis” instead of “vascular development”).

- Lines 130-133: This paragraph needs work: mention autocrine and paracrine Ang/Tie signaling, please add citations

We thank the Referee for highlighting the imprecisions and the lack of references in this section. The paragraph has been revised as suggested.

- Paragraph C. a. could be shortened and structured in a more stringent manner

This section has been revised as suggested.

- Lines 225 ff: DVIC should be replaced by DIC - the official term used in hematology. In terms of coagulopathies LIC needs to be mentioned as well and, specifically for KLA, the Kasabach-Merritt phenomenon is missing and should be added (as it precedes DIC in KLA).

We thank the Referee for pointing out this typo. We have revised the section accordingly.

As suggested, we have referred to LIC in the venous malformations section due to its localized pathophysiology; we have retained DIC in the KLA section, as its mechanisms remain incompletely understood and may involve systemic consumption.

However, we disagree with the Referee that Kasabach-Merritt phenomenon (KMP) classically occurs in KLA. Though we thoroughly reviewed the available literature, we could not find any reports of KMP in KLA. In contrast, KMP is a typical complication of vascular tumors such as Kaposiform hemangioendothelioma and tufted angioma, but not vascular malformations (Kelly 2010; Servattalab and Grenier 2025; McDaniel et al. 2023).

A detailed review on KLA published by McDaniel *et al.* highlights (McDaniel et al. 2023):

“Unlike Kasabach–Merritt phenomenon (KMP), which is associated with KHE and characterized by profound thrombocytopenia and hypofibrinogenemia, decreases in platelet levels are generally, but not always, more mild in cases of KLA.”

- Line 239 ff: Please use high-output cardiac failure throughout

We thank the Referee for this suggestion. We have modified the text accordingly.

- Lines 434 ff: It would be good to mention the relatively high organ toxicity of trametinib compared to dabrafenib

As suggested by the Referee, we have detailed the incidence of adverse events associated with trametinib in comparison to dabrafenib. These include high-grade dermatological toxicity, fever, fatigue, arterial hypertension, decreased left ventricular ejection fraction, venous thrombo-embolic events. We have underscored that the overall incidence of AEs associated with trametinib has been reported to be higher than that of dabrafenib-related AEs.

The trametinib section addressing adverse events now appears as follows:

“Unfortunately, the use of trametinib is limited by frequent AEs. In oncology trials, overall AEs incidence was 97% with trametinib monotherapy (including 36% grade ≥ 3), compared to 85% with dabrafenib monotherapy (Garutti et al. 2022). Dermatological and gastro-intestinal toxicities are the most frequent AEs of trametinib in cancer patients (Garutti et al. 2022). Indeed, trametinib was associated with acneiform (63% adults, 31% children) and maculopapular rash (42% adults, 39% children), alopecia and paronychia, diarrhea (73% adults, 46% children), and oral mucositis (35% adults, 31% children), often requiring dose adjustments in severe cases (Gershenson et al. 2022; Bouffet et al. 2023; Anforth et al. 2014). It was also associated with frequent fever (43% of adults, 31% of children) and fatigue (28% of adults, 39% of children) (Garutti et al. 2022; Bouffet et al. 2023). Similarly, in nonmalignant conditions, a meta-analysis of trametinib in pediatric neurofibromatosis reported paronychia in 61% of cases, rash in 26%, diarrhea in 17%, and mouth ulcers in 0.5%. (D.

Wang et al. 2022). Such AEs were also reported in patients with AVMs receiving trametinib (Nicholson et al. 2022; Seebauer et al. 2024). Trametinib was also linked to decreased left ventricular ejection fraction (LVEF), peripheral edema, and severe hypertension in 6-12% of adult patients (Banks et al. 2017; Mincu et al. 2019; Glen et al. 2022; Gershenson et al. 2022; Garutti et al. 2022; Bouffet et al. 2023). In cancer treatment, trametinib combined with dabrafenib was associated with a higher incidence of venous thromboembolism than dabrafenib alone (Mincu et al. 2019). Since VM patients are at risk of cardiac failure and thromboembolism, the use of MEK inhibitors requires careful consideration and close monitoring.“

- Lines 467 ff: Define bevacizumab first

Bevacizumab had already been defined in the previous paragraph called “VEGF inhibitors”. For clarity, we have moved the first occurrence of bevacizumab to the “anti-VEGF drugs” section.

- In the conclusion, the importance of combinational targeted strategies could be emphasized

We thank the Referee for the suggestion. We have modified the text accordingly:

“Notably, combination therapies, which aim to synergistically target dysregulated signaling pathways, may be promising strategies to enhance efficacy and overcome resistance, albeit with different tolerance profiles.”

- Figure 1: Please add a nucleus and I would suggest using the terms differentiation and proliferation instead; please add legend of signs used

Given the available evidence of the impact of PI3K and MAPK pathways on cell growth and form, we have kept the term “growth” in the end of the scheme. However, we have added the terms “differentiation” and “proliferation” as suggested, as well as the legend of the signs used in Figure 1.

Despite the Referee’s suggestion, we have decided not to include a nucleus in Figure 1 as it would not add any meaningful information to the scheme. A similar scheme was previously published in the review by (Queisser et al. 2021), without compromising the clarity of the figure.

- Figure 2C: The cMRI of the AVM could be more representative. For AVMs, it is generally most representative to show a TWIST Angio to appreciate the full extent of the lesion

We agree with the Referee’s observation and have revised the AVM panel. We have replaced it with angio CT-scan and arteriography pictures from a patient with a large thoracic AVM.

Besides, as the first and the second panel appeared quite similar, we have also replaced the lymphatic malformation image with a different view from the same patient to enhance the educational value.

- Genetics Table in Figure 2: This is not totally clear to me: e.g., AVMs can be isolated and syndromic and harbor far more mutations including HRAS, NRAS, ARAF and even PIK3CA

We agree with the Referee that the former Table 1 was somehow confusing. We have divided it into 2 parts for clarity, each addressing slow-flow vascular malformations and arteriovenous malformations. We have explicitly focused on the most frequently reported variant genes in each disease to avoid overcrowding. The table is purposely non exhaustive to improve clarity.

To our best knowledge, *PIK3CA* is not commonly associated with AVMs. However, *PTEN* variants are frequently involved in AVMs. We have therefore mentioned the most frequently reported genes (*KRAS*, *HRAS*, *NRAS*, *BRAF*, *MAP2K1* and *PTEN*) in the sporadic AVM panel.

Referee #3 (Remarks for Author):

The authors review the current knowledge of vascular malformations. Overall, this is a timely and interesting topic of research. While the authors have done a good job summarizing relevant information, some aspects require attention before publication. Particularly, there are many inconsistencies regarding references; authors should carefully revise those.

We thank the Referee for their positive feedback and agree that several inaccuracies remained in the reference list, due to citation indexing errors. These issues have been carefully reviewed and corrected. We apologize for the inconvenience.

Abstract: In line 27, include rapamycin, as extensive work on this drug and VMs has been done and published.

As requested (and discussed previously with Referee 2), we have modified the abstract accordingly (line 27):

“mTOR and PI3K α inhibitors such as sirolimus and alpelisib have shown promising efficacy in slow-flow VMs, while reports have suggested that MAPK inhibitors such as trametinib may improve arteriovenous malformations.”

In line 54, the authors include the incidence of VM and AVM but not of LM

We acknowledge that including epidemiological data regarding lymphatic malformations adds value to this section. The paragraph has been modified as follows:

“The prevalence of VMs is estimated around 1 in 1,000 live births with an annual incidence of 1 per 10,000 (Penington et al. 2023; Ryu et al. 2023), though these figures are likely imprecise due to the absence of prospective registries. Venous malformations, the most common subtype, are found in 4.5 out of 10,000 while lymphatic malformations account for 3.5 out of 10,000. In contrast, AVMs only account for 1 case in 10,000, but they are associated with the highest mortality rates (Penington et al. 2023; Ryu et al. 2023).”

Line 79, please put a reference at the end of the sentence (...additional events)

At the end of 79-82 sentences, add this publication Limaye N Nature Genetics 2009 (Vikkula lab)

As requested, we have added references to reports of second-hit mutations in vascular malformations, including Limaye et al.:

“Moreover, familial VMs follow a paradominant inheritance pattern, where inherited variants frequently combine with somatic second-hit mutations to drive the phenotype (Seront et al. 1993). Consequently, germline variants usually lead to multifocal asymmetrical malformations depending on localized additional events, as shown in various familial vascular malformation syndromes (Limaye et al. 2009; Revencu et al. 2013; Amyere et al. 2013; Macmurdo et al. 2016; Hill et al. 2021; DeBose-Scarlett et al. 2024; Castillo et al. 2025).”

In line 91, what do the authors mean with VM syndromes? And why KRAS-related AVMs are considered so?

We thank the Referee for highlighting this typo. 'Syndromes' had been left out by mistake during the writing process. We have corrected the sentence as follows:

"Numerous other genes were then reported as causative in various VMs, such as AK strain Transforming gene (*AKT*, in Proteus syndrome), *PIK3CA* (in *PIK3CA*-related overgrowth syndromes, PROS) or, more recently, Kirstein RA^t Sarcoma viral oncogene homolog (*KRAS*, in brain AVMs) (Lindhurst et al. 2011; Kurek et al. 2012; Nikolaev et al. 2018)."

Please rewrite the sentence in line 117, it is not clear.

As suggested, we have corrected the sentence as follows:

"On the other hand, AVM variants most frequently involve genes of the RAS-MAPK pathway"

Line 125, please consistently use the term PROS in the review, instead of other clinical syndromes (KTS). If the specific word of KTS wants to be referred to, explain is a subtype of PROS.

As suggested, we have corrected the sentence as follows:

"VMs can also occur in complex syndromes such as PROS, which may involve capillary, lymphatic, or venous malformations and asymmetrical overgrowth"

Line 130-131, include references at the end of the sentence

As requested, we have added references to articles related to Angiopoietins and the TIE2 pathway:

"The TIE2 receptor (encoded by *TEK*) is expressed at the surface of ECs and binds angiopoietins (ANG), a family of endothelial growth factors (**Figure 1**). Angiopoietin 1 (ANG1) is secreted by perivascular cells such as vascular smooth muscle cells (VSMCs) and promotes vascular maturation and stability (Augustin et al. 2009; Thomas and Augustin 2009), while angiopoietin 2 (ANG2) signals in ECs in an autocrine fashion (Yuan et al. 2009; Korhonen et al. 2016). Activation of TIE2 initiates the PI3K α pathway, which is critical for vascular maturation, remodeling, and angiogenesis (Seront et al. 1993)."

In line 141, please change the word recruit to activate. PIP3 does not recruit mTORC2...

As suggested, we have changed the word 'recruit' to 'activate'.

In line 152, Makinen et al. is not the correct reference to the above statement. Please change it. Also, I am not sure if the link between ANG-2 and PDGF with disruption of VSMCs has been shown. If so, include a reference

We thank the Referee for highlighting these mistakes. We have restructured the whole section and removed the sentence relating to disruption of VSMCs.

At the end of line 150, add the preprint Sabata et al.

As suggested by the Referee, we have added a reference to the preprint by Sabata et al:

“Dysregulation of the PI3K α pathway has morphological and functional consequences in ECs. Gain-of-function variants in *TEK* and *PIK3CA* lead to excessive PI3K α , AKT and mTOR signaling, driving senescence, increased vessel permeability, altered arteriovenous identity, and aberrant angiogenic potential (Sabata et al. 2025; Bloom et al. 2023).”

In line 157, the reference Luks et al is incorrect. The authors should replace it with Castillo et al. 2016, Castel et al. 2016, and Limaye et al. 2015.

We apologize for the indexation errors. As requested, we have carefully revised the citations and replaced the preexisting reference with those suggested:

“Further studies revealed somatic GoF mutations in *TEK* (mainly the L914F variant) in over 60% sporadic venous malformations, and in *PIK3CA* (mainly hotspot variants E542K, E545K, and H1047R) in 20% (Castillo et al. 2016; Castel et al. 2016a; Limaye et al. 2015).”

In line 158, the reference Kobialka et al is incorrect. Please cite a LM paper here

Here again, we apologize for the inconvenience. We have added the following references to the section: (Osborn et al. 2015; Luks et al. 2015).

In line 163, Luks et al is wrong here, add instead Castillo et al. 2025

As suggested, we have added the following references to reports of *PTEN* somatic variants in PHTS: (Liaw et al. 1997; Castillo et al. 2025; Salo-Mullen et al. 2014; Hendricks et al. 2022).

Add a reference which refers to the angiogenic secretome in line 166

As requested, we have added the following reference that mentions ECs' angiogenic secretome: (Castel et al. 2016b).

In line 168, the reference Graupera et al incorrect (this is a LOF model). Please change it to Hare et al. 2015 (from the Wayne Phillips lab).

We apologize for this mistake. As suggested, we have modified the reference and cited (Hare et al. 2015).

Please re-write the sentence in line 177; "The CCM complex....). While the authors summarize the scheme of that paper (Ren et al.)... Currently, it is not fully elucidated how the different signalling components interact. The way that the sentence is written implies this is indeed what happens, and at this stage, it is just a proposition based on the data of Ren et al. Please turn down the tone at least

We acknowledge that the sentence seemed to imply that those assumptions were supported by strong evidence. We have simplified the sentence by removing the reference to KLFs to ease the understanding of this section. We have also turned down the tone of the conclusion to underscore the hypothetical nature of such assumptions:

"Cerebral cavernous malformations (CCMs) are venous malformations of the brain at risk of bleeding, resulting in epilepsy, focal neurological deficits, or headaches (Snellings et al. 2021). Familial CCM syndromes result from germline mutations in one of the three *CCM* genes (*CCM1*, 2 and 3) which encode components of the CCM complex. This complex acts as an inhibitor of MEKK3 (encoded by *MAP3K3*), involved in the MAPK pathway (**Figure 1**) (Zhou et al. 2016). LoF of CCM complex results in increased MEKK3 signaling and activation of the mTOR pathway (Ren et al. 2021; Zhou et al. 2016).

CCMs development often follows a two-step mechanism involving both the CCM-MAP3K3 axis and the PI3K pathway. In familial CCMs, germline mutations in the *CCM* genes are frequently associated with an additional post-zygotic variant in *PIK3CA* (Ren et al. 2021). Similarly, sporadic CCMs may involve postzygotic variants in *PIK3CA* and additional variants in the *CCM* genes or in *MAP3K3*. Interestingly, sporadic CCMs often arise near developmental venous anomalies (DVA), a type of brain venous malformation caused by somatic *PIK3CA* variants, which may serve as a primer for CCMs (Petersen et al. 2010; Snellings et al. 2022). As both *PIK3CA* GoF and *CCM* LoF converge towards the activation of mTORC1, they probably synergistically contribute to CCM growth (Ren et al. 2021)."

The sentence in 183 is incorrect. There is a second hit in some cases, but not in all. Please correct

We agree with the Referee that second hits are not reported in all CCMs, and we have therefore corrected the sentence as follows:

"CCMs development frequently follows a two-step mechanism involving the CCM-MAP3K3 axis and the PI3K pathway. Indeed, familial CCMs involve germline mutations in the *CCM* genes and frequently exhibit an additional post-zygotic variant in *PIK3CA* (Ren et al. 2021). Similarly, sporadic CCMs may involve postzygotic variants in *PIK3CA* and additional variants in the *CCM* genes or in *MAP3K3*."

Something is missing at the end of the sentence starting in line 206...

We apologize for the inconvenience. We have corrected the sentence as follows for clarity:

“Most frequently in AVMs, mosaic GoF variants are found in *RAS*, *BRAF*, *MAP2K1* (encoding for MEK1), *ARAF* or *SOS1* (Figure 2).”

For the *NRAS* Q61R, cite Manevitz-Mendelson et al. (Angiogenesis 2018), as this is the first manuscript in which the mutation is identified in KLA.

In the reference cited by the Referee (Manevitz-Mendelson et al. 2018), the authors state:

“Based on the radiologic and histological findings, generalized lymphatic anomaly (GLA) was diagnosed. Kaposiform lymphangiomatosis (KLA), a newly appreciated subtype of GLA, was also considered due to the aggressive clinical behavior and the histological findings of focally dispersed spindle-like cells [...], as described in this entity. However, lack of cutaneous involvement and consumptive coagulopathy were against this disorder.”

Therefore, given the complexity of such lymphatic anomalies and the overlap between the different syndromes making them difficult to apprehend, we have not added the reference to the manuscript to avoid misleading the readers.

In line 263, please use activate instead of recruit. Also, PLC β activates RAS-MAPK but not PI3K.

We have modified the sentence accordingly:

“G α q and G α 11, encoded by guanine nucleotide-binding protein subunit alpha q (*GNAQ*) and 11 (*GNA11*), activate phospholipase C- β (PLC β), which in turn activates the RAS-MAPK pathway”

In line 269, R183Q does not result in constitutive activation of *GNAQ*; what it does is sustain the GTP loading, which sustains the activation longer. Correct.

We thank the Referee for this constructive comment. We acknowledge that the former version was oversimplified and inaccurate. The sentence has been modified accordingly:

“The *GNAQ* R183Q variant reduces G α q affinity for GDP and sustains the activation of G α q”

In Shirley et al., the authors show that pAkt is not altered upon expressing R183Q in ECs. Thus, the sentence in line 269 is incorrect: *GNAQ* does not dysregulate the PI3K pathways, similarly to *PIK3CA* mutations.

We thank the Referee for pointing out this mistake. The sentence has been corrected as follows:

“Consequently, mutant *GNAQ* may impair ECs response to blood flow, dysregulate the MAPK pathway, and impair EC differentiation, leading to capillary malformations.”

In line 273, the GNAQ Q209L variant is not the only causing congenital hemangioma.

We agree with the Referee that the Q209L variant is not the only variant found in hemangiomas. However, this is not what we implied; we aimed to underline the variant-selectivity of vascular anomalies (Q209L in hemangiomas and R183Q in VMs), but not to suggest any exclusivity between a given variant and hemangiomas. The paragraph has therefore been modified as follows for clarity:

“Interestingly, variants in *GNAQ* are also found in vascular tumors such as congenital hemangiomas (Ayturk et al. 2016). However, *GNAQ* variants associated with vascular tumors seem to exhibit stronger impact on the G α pathway. Indeed, the most prevalent variant in hemangiomas, Q209L, shows higher activation of G α signaling compared to R183Q, which is predominant in VMs (Ayturk et al. 2016; L. Martins et al. 2017; Galeffi et al. 2022). Therefore, the apparent preference for specific *GNAQ* variants in different types of vascular anomalies may reflect their distinct functional effects.”

In line 296, what does it mean suspensive?

To improve clarity, we have replaced ‘suspensive’ by ‘temporary’:

“However, the efficacy of alpelisib was only temporary as disease recurred upon cessation of treatment (Venot et al. 2018b).”

In line 303, mutations in PIK3R1 have been published

(<https://www.sciencedirect.com/science/article/pii/S1769721222001719?via%3Dihub>)

We agree with the Referee. The reference was already included in the main text – but misplaced after the mention of *TEK* variants. For clarity, we have moved it to the end of the sentence:

“Interestingly, it also showed effectiveness in VMs caused by variants in genes encoding regulators of PI3K α such as *TEK* (Remy et al. 2022; Sterba et al. 2023b; Zerbib et al. 2024) and *PIK3R1* (Schönewolf-Greulich et al. 2022; Morin et al. 2025).”

I would suggest using PIK3CA instead of p110a in line 302. This would help clinicians to read through. Also, be consistent with p110a or PI3K α (both are fine as long as they are defined) but I would suggest not mixing (e.g, line 302, 309)

We agree with the Referee that using consistent nomenclature would improve the readability of the manuscript. As suggested, we have replaced “p110 α ” with “PI3K α ” throughout the text for clarity.

The authors should be clearer about the side effects and growth effects of Alpelisib. It is now well known that some patients, including PROS patients, cannot tolerate this treatment for a long time and that the growth effects are an issue.

As requested, and in line with our previous discussion with Referee 2, we have rephrased the paragraph to more accurately reflect the recent concerns raised by the international community:

“Though no warning has been issued by health authorities nor by most of the available literature so far (Pagliuzzi et al. 2021; Garreta Fontelles et al. 2022; Schönewolf-Greulich et al. 2022; Canaud et al. 2023; Kolitz et al. 2022), one case of growth delay was recently reported in a pediatric patient treated with alpelisib (Etingin et al. 2025). The occurrence of growth delay may be dose-dependent, considering that it was observed in the child with the highest drug exposure among 8 treated patients (Etingin et al. 2025). Until long-term follow-up data in large therapeutic cohorts are available, careful monitoring is therefore required in children treated with alpelisib.”

In line 328, change Mirasentib for Akt inhibitors and include capivasertib since the last publication by Castillo et al. shows efficacy in PHTS-related vascular malformations in preclinical models.

We thank the Referee for this constructive comment. As suggested, we have added the following sentence to the manuscript:

“More recently, capivasertib, another AKT inhibitor, showed comparable efficacy as observed with sirolimus at reducing mTORC1 signaling and vascular hyperplasia in the postnatal retina in a preclinical model of PTEN-related vascular malformations (Castillo et al. 2025).”

In line 333, the authors refer to preclinical studies, but the reference pertains to clinical studies.

We thank the Referee for highlighting this mistake. The sentence has been corrected as follows:

“Clinical studies reported its efficacy in patients with Proteus syndrome and (Leoni et al. 2019; Ours et al. 2021; Forde et al. 2021).”

Given that Everolimus is the best-studied drug in these conditions, the review would benefit from including some discussions about on- and off-treatment, the impact of different doses, and whether the AEs are dose-dependent.

We agree with the Referee that including supplementary pharmacological data on everolimus adds value to the section relating to rapalogs. As suggested, we have revised the paragraph and included sections on AEs across diseases (transplantation, cancer and rare diseases), the impact of low-dose versus high-dose regimens, and the dose-dependency of most AEs reported.

In line 365, include this information: Sirolimus has proven preclinical and clinical therapeutic efficacy in PHTS-related vascular malformations

We agree with the Referee that this information was missing and adds value to this section. We have added the following sentence:

“Sirolimus also showed preclinical and clinical efficacy in PTEN-related VMs, including AVMs (Zabeida et al. 2024; Castillo et al. 2025).”

In line 434, the authors state that trametinib shows frequent side effects. How frequent is that compared to Alpelisib? Can authors also provide information on the frequency of Alpelisib?

We agree with the Referee that precise epidemiological data relating to alpelisib-related adverse events was missing. We have therefore modified the section as follows:

“In the EPIK-P1 trial, drug-related AEs occurred in 38.6% of patients, notably hyperglycemia (27.8% of adults, 5.1% of children, grade ≤ 2) and aphthous ulcers (16.7% and 7.7%, respectively) (Canaud et al. 2023). In the EPIK-P2 trial, drug-related AEs occurred in 57.4% of adults receiving alpelisib (versus 44.4% in the placebo group) and 33.9% of children (versus 32.1%) (Canaud et al. 2024, 2). Though no warning has been issued by health authorities nor by most of the available literature so far, cases of growth delay were recently reported in a few pediatric patients treated with alpelisib (Sterba et al. 2023b; Etingin et al. 2025; Pagliuzzi et al. 2021; Garreta Fontelles et al. 2022; Schönewolf-Greulich et al. 2022; Canaud et al. 2023; Koltz et al. 2022). Growth restriction was observed in the youngest patients receiving higher weight-adjusted doses (3.6-4.8 mg/kg/day), consistent with a dose-dependent effect due to increased drug exposure (Etingin et al. 2025). We did not observe such complications in the infants we treated with lower weight-adjusted doses (2.8 and 3.1 mg/kg/day), suggesting that growth delay may occur at doses exceeding 3 mg/kg/day (Morin et al. 2022). As highlighted in the recent report by Remy et al., determining the optimal weight-adjusted dosage of alpelisib in the pediatric population based on pharmacokinetic data is therefore essential (Remy et al. 2025). Until long-term follow-up data from large therapeutic cohorts become available, pediatric patients receiving alpelisib should be closely monitored.”

We have also underlined alpelisib-related AEs reported in previous oncology studies (SOLAR study):

“In the phase III SOLAR study assessing alpelisib (300mg/d) in 284 patients with *PIK3CA*-mutated, hormone receptor-positive breast cancer, hyperglycemia (64% patients), diarrhea, nausea, decreased appetite and rash were the most frequent all-grade AEs (André et al. 2019). Similar AEs had been previously reported in an oncology dose escalation study (up to 450mg/d alpelisib) and in a phase 1b study, where hyperglycemia and rash were reported to be the most common dose-dependent, alpelisib-related AEs. Hyperglycemia was manageable by dose interruption or metformin in most patients, while rash occurred within 2 weeks of treatment and was manageable by antihistamines and steroids (Juric et al. 2018; 2019).”

Finally, we have detailed the frequency of trametinib-related AEs in the corresponding section:

“Unfortunately, the use of trametinib is limited by frequent AEs. In oncology trials, overall AEs incidence was 97% with trametinib monotherapy (including 36% grade ≥ 3), compared to 85% with dabrafenib monotherapy (Garutti et al. 2022). Dermatological and gastro-intestinal toxicities are the most frequent AEs of trametinib in cancer patients (Garutti et al. 2022). Indeed, trametinib was associated with acneiform (63% adults, 31% children) and maculopapular rash (42% adults, 39% children), alopecia and paronychia, diarrhea (73% adults, 46% children), and oral mucositis (35% adults, 31% children), often requiring dose adjustments in severe cases (Gershenson et al. 2022; Bouffet et al. 2023; Anforth et al. 2014). It was also associated with frequent fever (43% of adults, 31% of children) and fatigue (28% of adults, 39% of children) (Garutti et al. 2022; Bouffet et al. 2023). Similarly, in nonmalignant conditions, a meta-analysis of trametinib in pediatric neurofibromatosis reported paronychia in 61% of cases, rash in 26%, diarrhea in 17%, and mouth ulcers in 0.5%. (D. Wang et al. 2022). Such AEs were also reported in patients with AVMs receiving trametinib (Nicholson

et al. 2022; Seebauer et al. 2024). Trametinib was also linked to decreased left ventricular ejection fraction (LVEF), peripheral edema, and severe hypertension in 6-12% of adult patients (Banks et al. 2017; Mincu et al. 2019; Glen et al. 2022; Gershenson et al. 2022; Garutti et al. 2022; Bouffet et al. 2023). In cancer treatment, trametinib combined with dabrafenib was associated with a higher incidence of venous thromboembolism than dabrafenib alone (Mincu et al. 2019). Since VM patients are at risk of cardiac failure and thromboembolism, the use of MEK inhibitors requires careful consideration and close monitoring.”

In line 459, I do not understand what the authors mean by the sentence! Though VEGF signals.... Please explain better. The whole section on VEGF needs some attention.

We acknowledge that the original sentence was misleading and imprecise. We have corrected it as follows:

“Moreover, elevated VEGF levels have been observed in the serum and nasal mucosa of HHT patients (Mansur and Radovanovic 2023). Interestingly, VEGF seems to fuel the progression of AVMs: intracranial injections of VEGF increased the incidence of brain hemorrhage, vascular leakage and mortality rates in an *ALK1*-deficient HHT mouse model (Cheng et al. 2019). Therefore, VEGF may be a relevant therapeutic target in germline AVM syndromes.”

We have also carefully reviewed the whole VEGF section and corrected it where needed to improve clarity.

What do the authors mean by "**indirect inhibitor of VEGF**" in line 490?

We thank the Referee for highlighting this imprecision. As requested, we have corrected the sentence for clarity:

“More recently, thalidomide was reported to reduce bleeding in germline and sporadic AVMs. In vitro studies reported that thalidomide represses VEGF synthesis and depletes VEGFRs in human ECs (Yabu et al. 2005; Komorowski et al. 2006).”

In line 515, include the use the **notion that Aflibercept has also been tested for HHT** (<https://www.mdpi.com/1648-9144/59/9/1533>)

As suggested by the referee, we have added the following section addressing the use of aflibercept in a patient with HHT:

“Additionally, aflibercept, a soluble VEGF decoy receptor, improved hemoglobin levels and reduced bleeding and transfusion needs after 6 months in an HHT patient with bevacizumab-refractory gastrointestinal bleeding (Villanueva et al. 2023). Therefore, aflibercept may offer a promising alternative for patients unresponsive or ineligible to bevacizumab.”

References – Response to Referees

- Adams, Denise M., Cameron C. Trenor, Adrienne M. Hammill, et al. 2016. “Efficacy and Safety of Sirolimus in the Treatment of Complicated Vascular Anomalies.” *Pediatrics* 137 (2): e20153257. <https://doi.org/10.1542/peds.2015-3257>.
- Amyere, Mustapha, Virginie Aerts, Pascal Brouillard, et al. 2013. “Somatic Uniparental Isodisomy Explains Multifocality of Glomuvenous Malformations.” *American Journal of Human Genetics* 92 (2): 188–96. <https://doi.org/10.1016/j.ajhg.2012.12.017>.
- André, Fabrice, Eva Ciruelos, Gabor Rubovszky, et al. 2019. “Alpelisib for PIK3CA-Mutated, Hormone Receptor-Positive Advanced Breast Cancer.” *The New England Journal of Medicine* 380 (20): 1929–40. <https://doi.org/10.1056/NEJMoa1813904>.
- Anforth, Rachael, Michael Liu, Bao Nguyen, et al. 2014. “Acneiform Eruptions: A Common Cutaneous Toxicity of the MEK Inhibitor Trametinib.” *The Australasian Journal of Dermatology* 55 (4): 250–54. <https://doi.org/10.1111/ajd.12124>.
- Augustin, Hellmut G., Gou Young Koh, Gavin Thurston, and Kari Alitalo. 2009. “Control of Vascular Morphogenesis and Homeostasis through the Angiopoietin-Tie System.” *Nature Reviews. Molecular Cell Biology* 10 (3): 165–77. <https://doi.org/10.1038/nrm2639>.
- Ayturk, Ugur M., Javier A. Couto, Steven Hann, et al. 2016. “Somatic Activating Mutations in GNAQ and GNA11 Are Associated with Congenital Hemangioma.” *American Journal of Human Genetics* 98 (4): 789–95. <https://doi.org/10.1016/j.ajhg.2016.03.009>.
- Banks, Mary, Karen Crowell, Amber Proctor, and Brian C. Jensen. 2017. “Cardiovascular Effects of the MEK Inhibitor, Trametinib: A Case Report, Literature Review, and Consideration of Mechanism.” *Cardiovascular Toxicology* 17 (4): 487–93. <https://doi.org/10.1007/s12012-017-9425-z>.
- Bissler, John J., J. Chris Kingswood, Elzbieta Radzikowska, et al. 2017. “Everolimus Long-Term Use in Patients with Tuberous Sclerosis Complex: Four-Year Update of the EXIST-2 Study.” *PloS One* 12 (8): e0180939. <https://doi.org/10.1371/journal.pone.0180939>.
- Bissler, John J., J. Christopher Kingswood, Elzbieta Radzikowska, et al. 2013. “Everolimus for Angiomyolipoma Associated with Tuberous Sclerosis Complex or Sporadic Lymphangiomyomatosis (EXIST-2): A Multicentre, Randomised, Double-Blind, Placebo-Controlled Trial.” *The Lancet* 381 (9869): 817–24. [https://doi.org/10.1016/S0140-6736\(12\)61767-X](https://doi.org/10.1016/S0140-6736(12)61767-X).
- Bloom, Samuel I., Md Torikul Islam, Lisa A. Lesniewski, and Anthony J. Donato. 2023. “Mechanisms and Consequences of Endothelial Cell Senescence.” *Nature Reviews Cardiology* 20 (1): 38–51. <https://doi.org/10.1038/s41569-022-00739-0>.
- Bouffet, Eric, Birgit Georger, Christopher Moertel, et al. 2023. “Efficacy and Safety of Trametinib Monotherapy or in Combination With Dabrafenib in Pediatric BRAF V600-Mutant Low-Grade Glioma.” *Journal of Clinical Oncology* 41 (3): 664–74. <https://doi.org/10.1200/JCO.22.01000>.
- Canaud, Guillaume, Juan Carlos Lopez Gutierrez, Alan D. Irvine, et al. 2023. “Alpelisib for Treatment of Patients with PIK3CA-Related Overgrowth Spectrum (PROS).” *Genetics in Medicine* 25 (12): 100969. <https://doi.org/10.1016/j.gim.2023.100969>.
- Canaud, Guillaume, Juan Carlos López-Gutiérrez, Adrienne M Hammill, et al. 2024. “Epik-P2: A Phase 2 Study of Alpelisib (ALP) in Pediatric and Adult Patients (Pts) with PIK3CA-Related Overgrowth Spectrum (PROS).” *Blood* 144 (Supplement 1): 5512. <https://doi.org/10.1182/blood-2024-198497>.
- Castel, Pau, F. Javier Carmona, Joaquim Grego-Bessa, et al. 2016a. “Somatic PIK3CA Mutations as a Driver of Sporadic Venous Malformations.” *Science Translational Medicine* 8 (332): 332ra42. <https://doi.org/10.1126/scitranslmed.aaf1164>.
- Castel, Pau, F. Javier Carmona, Joaquim Grego-Bessa, et al. 2016b. “Somatic PIK3CA Mutations as a Driver of Sporadic Venous Malformations.” *Science Translational Medicine* 8 (332): 332ra42–332ra42. <https://doi.org/10.1126/scitranslmed.aaf1164>.
- Castillo, Sandra D., Xabier Perosanz, Andrew K. Ressler, et al. 2025. “Somatic Uniparental Disomy of PTEN in Endothelial Cells Causes Vascular Malformations in Patients with PTEN Hamartoma Tumor Syndrome.” *Cancer Discovery*, May 14, OF1–13. <https://doi.org/10.1158/2159-8290.CD-24-0807>.

- Castillo, Sandra D., Elena Tzouanacou, May Zaw-Thin, et al. 2016. "Somatic Activating Mutations in Pik3ca Cause Sporadic Venous Malformations in Mice and Humans." *Science Translational Medicine* 8 (332): 332ra43. <https://doi.org/10.1126/scitranslmed.aad9982>.
- Cheng, Philip, Li Ma, Sonali Shaligram, et al. 2019. "Effect of Elevation of Vascular Endothelial Growth Factor Level on Exacerbation of Hemorrhage in Mouse Brain Arteriovenous Malformation." *Journal of Neurosurgery* 132 (5): 1566–73. <https://doi.org/10.3171/2019.1.JNS183112>.
- "Classification | International Society for the Study of Vascular Anomalies." n.d. Accessed July 27, 2024. <https://www.issva.org/classification>.
- Cossio, María-Laura, Josefina Rodríguez, Juan Carlos Flores, et al. 2024. "Four-Month-Old with Severe PIK3CA-Related Overgrowth Spectrum Disorder Successfully Treated with Alpelisb." *Pediatric Dermatology* 41 (4): 714–17. <https://doi.org/10.1111/pde.15582>.
- DeBose-Scarlett, Evon, Andrew K. Ressler, Carol J. Gallione, et al. 2024. "Somatic Mutations in Arteriovenous Malformations in Hereditary Hemorrhagic Telangiectasia Support a Bi-Allelic Two-Hit Mutation Mechanism of Pathogenesis." *The American Journal of Human Genetics* 111 (10): 2283–98. <https://doi.org/10.1016/j.ajhg.2024.08.020>.
- Etingin, Albert, Remy ,Amandine, Sonea ,Thomas, et al. 2025. "Alpelisib in Pediatric PIK3CA- and TIE-2–Mutant Vascular Anomalies: A Case Series on Safety, Efficacy, and Drug Exposure." *Pediatric Hematology and Oncology* 42 (4): 228–41. <https://doi.org/10.1080/08880018.2025.2498660>.
- Fakih, Marwan G., Scott Kopetz, Yasutoshi Kuboki, et al. 2022. "Sotorasib for Previously Treated Colorectal Cancers with KRASG12C Mutation (CodeBreak100): A Prespecified Analysis of a Single-Arm, Phase 2 Trial." *The Lancet Oncology* 23 (1): 115–24. [https://doi.org/10.1016/S1470-2045\(21\)00605-7](https://doi.org/10.1016/S1470-2045(21)00605-7).
- Fakih, Marwan G., Lisa Salvatore, Taito Esaki, et al. 2023. "Sotorasib plus Panitumumab in Refractory Colorectal Cancer with Mutated KRAS G12C." *New England Journal of Medicine* 389 (23): 2125–39. <https://doi.org/10.1056/NEJMoa2308795>.
- Forde, Karina, Nicoletta Resta, Carlotta Ranieri, et al. 2021. "Clinical Experience with the AKT1 Inhibitor Miransertib in Two Children with PIK3CA-Related Overgrowth Syndrome." *Orphanet Journal of Rare Diseases* 16 (February). <https://doi.org/10.1186/s13023-021-01745-0>.
- Fraissenon, Antoine, Charles Bayard, Gabriel Morin, et al. 2024. "Sotorasib for Vascular Malformations Associated with KRAS G12C Mutation." *New England Journal of Medicine* 391 (4): 334–42. <https://doi.org/10.1056/NEJMoa2309160>.
- Freixo, Cristiana, Vítor Ferreira, Joana Martins, et al. 2020. "Efficacy and Safety of Sirolimus in the Treatment of Vascular Anomalies: A Systematic Review." *Journal of Vascular Surgery* 71 (1): 318–27. <https://doi.org/10.1016/j.jvs.2019.06.217>.
- Gabeff, Romain, Olivia Boccara, Véronique Soupre, et al. 2019. "Efficacy and Tolerance of Sirolimus (Rapamycin) for Extracranial Arteriovenous Malformations in Children and Adults." *Acta Dermato-Venereologica* 99 (12): 1105–9. <https://doi.org/10.2340/00015555-3273>.
- Galeffi, F., D. A. Snellings, S. E. Wetzel-Strong, et al. 2022. "A Novel Somatic Mutation in GNAQ in a Capillary Malformation Provides Insight into Molecular Pathogenesis." *Angiogenesis* 25 (4): 493–502. <https://doi.org/10.1007/s10456-022-09841-w>.
- Garreta Fontelles, Gemma, Júlia Pardo Pastor, and Carme Grande Moreillo. 2022. "Alpelisib to Treat CLOVES Syndrome, a Member of the PIK3CA-Related Overgrowth Syndrome Spectrum." *British Journal of Clinical Pharmacology* 88 (8): 3891–95. <https://doi.org/10.1111/bcp.15270>.
- Garutti, Mattia, Melissa Bergnach, Jerry Polesel, Lorenza Palmero, Maria Antonietta Pizzichetta, and Fabio Puglisi. 2022. "BRAF and MEK Inhibitors and Their Toxicities: A Meta-Analysis." *Cancers* 15 (1): 141. <https://doi.org/10.3390/cancers15010141>.
- Gershenson, David M., Austin Miller, William E. Brady, et al. 2022. "Trametinib versus Standard of Care in Patients with Recurrent Low-Grade Serous Ovarian Cancer (GOG 281/LOGS): An International, Randomised, Open-Label, Multicentre, Phase 2/3 Trial." *The Lancet* 399 (10324): 541–53. [https://doi.org/10.1016/S0140-6736\(21\)02175-9](https://doi.org/10.1016/S0140-6736(21)02175-9).

Glen, Claire, Yun Yi Tan, Ashita Waterston, et al. 2022. "Mechanistic and Clinical Overview Cardiovascular Toxicity of BRAF and MEK Inhibitors." *JACC: CardioOncology* 4 (1): 1–18. <https://doi.org/10.1016/j.jacc.2022.01.096>.

Happle, R. 1978. "[Genetic interpretation of linear skin abnormalities]." *Der Hautarzt; Zeitschrift Fur Dermatologie, Venerologie, Und Verwandte Gebiete* 29 (7): 357–63.

Hare, Lauren M., Quenten Schwarz, Sophie Wiszniak, et al. 2015. "Heterozygous Expression of the Oncogenic *Pik3ca*(H1047R) Mutation during Murine Development Results in Fatal Embryonic and Extraembryonic Defects." *Developmental Biology* 404 (1): 14–26. <https://doi.org/10.1016/j.ydbio.2015.04.022>.

Hendricks, Linda A. J., Janneke Schuurs-Hoeijmakers, Isabel Spier, et al. 2022. "Catch Them If You Are Aware: *PTEN* Postzygotic Mosaicism in Clinically Suspicious Patients with *PTEN* Hamartoma Tumour Syndrome and Literature Review." *European Journal of Medical Genetics* 65 (7): 104533. <https://doi.org/10.1016/j.ejmg.2022.104533>.

Hill, Lauren R. S., Jessica Duis, Ann M. Kulungowski, Aparna Annam, Bradford Siegele, and Taizo A. Nakano. 2021. "A Challenging Diagnosis: *PTEN* Hamartoma Tumor Syndrome Presenting as Isolated Soft-Tissue Vascular Anomalies." *Journal of Vascular Anomalies* 2 (2): e011. <https://doi.org/10.1097/JOVA.0000000000000011>.

Hong, David S., Marwan G. Fakih, John H. Strickler, et al. 2020. "KRASG12C Inhibition with Sotorasib in Advanced Solid Tumors." *New England Journal of Medicine* 383 (13): 1207–17. <https://doi.org/10.1056/NEJMoa1917239>.

Johnston, Olwyn, Caren L. Rose, Angela C. Webster, and John S. Gill. 2008. "Sirolimus Is Associated with New-Onset Diabetes in Kidney Transplant Recipients." *Journal of the American Society of Nephrology: JASN* 19 (7): 1411–18. <https://doi.org/10.1681/ASN.2007111202>.

Juric, Dejan, Filip Janku, Jordi Rodón, et al. 2019. "Alpelisib Plus Fulvestrant in PIK3CA-Altered and PIK3CA-Wild-Type Estrogen Receptor-Positive Advanced Breast Cancer." *JAMA Oncology* 5 (2): e184475. <https://doi.org/10.1001/jamaoncol.2018.4475>.

Juric, Dejan, Jordi Rodon, Josep Tabernero, et al. 2018. "Phosphatidylinositol 3-Kinase α -Selective Inhibition With Alpelisib (BYL719) in PIK3CA-Altered Solid Tumors: Results From the First-in-Human Study." *Journal of Clinical Oncology* 36 (13): 1291–99. <https://doi.org/10.1200/JCO.2017.72.7107>.

Kaplan, Bruce, Yasir Qazi, and Jason R. Wellen. 2014. "Strategies for the Management of Adverse Events Associated with mTOR Inhibitors." *Transplantation Reviews* 28 (3): 126–33. <https://doi.org/10.1016/j.trre.2014.03.002>.

Kelly, Michael. 2010. "Kasabach-Merritt Phenomenon." *Pediatric Clinics of North America*, Birthmarks of Medical Significance, vol. 57 (5): 1085–89. <https://doi.org/10.1016/j.pcl.2010.07.006>.

Kolitz, Elysha, Neil J. Fernandes, Nnenna G. Agim, and Kathleen Ludwigl. 2022. "Response to Alpelisib in Clinically Distinct Pediatric Patients With PIK3CA-Related Disorders." *Journal of Pediatric Hematology/Oncology* 44 (8): 482. <https://doi.org/10.1097/MPH.0000000000002418>.

Komorowski, Jan, Hanna Jerczyńska, Agnieszka Siejka, et al. 2006. "Effect of Thalidomide Affecting VEGF Secretion, Cell Migration, Adhesion and Capillary Tube Formation of Human Endothelial EA.Hy 926 Cells." *Life Sciences* 78 (22): 2558–63. <https://doi.org/10.1016/j.lfs.2005.10.016>.

Korhonen, Emilia A., Anita Lampinen, Hemant Giri, et al. 2016. "Tie1 Controls Angiopoietin Function in Vascular Remodeling and Inflammation." *The Journal of Clinical Investigation* 126 (9): 3495–510. <https://doi.org/10.1172/JCI84923>.

Krueger, Darcy A., Marguerite M. Care, Katherine Holland, et al. 2010. "Everolimus for Subependymal Giant-Cell Astrocytomas in Tuberous Sclerosis." *New England Journal of Medicine* 363 (19): 1801–11. <https://doi.org/10.1056/NEJMoa1001671>.

Kurek, Kyle C., Valerie L. Luks, Ugur M. Ayturk, et al. 2012. "Somatic Mosaic Activating Mutations in PIK3CA Cause CLOVES Syndrome." *The American Journal of Human Genetics* 90 (6): 1108–15. <https://doi.org/10.1016/j.ajhg.2012.05.006>.

- Langen, Adrianus Johannes de, Melissa L Johnson, Julien Mazieres, et al. 2023. "Sotorasib versus Docetaxel for Previously Treated Non-Small-Cell Lung Cancer with *KRAS*G12C Mutation: A Randomised, Open-Label, Phase 3 Trial." *The Lancet* 401 (10378): 733–46. [https://doi.org/10.1016/S0140-6736\(23\)00221-0](https://doi.org/10.1016/S0140-6736(23)00221-0).
- Leoni, Chiara, Giuseppe Gullo, Nicoletta Resta, et al. 2019. "First Evidence of a Therapeutic Effect of Miransertib in a Teenager with Proteus Syndrome and Ovarian Carcinoma." *American Journal of Medical Genetics Part A* 179 (7): 1319–24. <https://doi.org/10.1002/ajmg.a.61160>.
- Liaw, Danny, Debbie J. Marsh, Jing Li, et al. 1997. "Germline Mutations of the PTEN Gene in Cowden Disease, an Inherited Breast and Thyroid Cancer Syndrome." *Nature Genetics* 16 (1): 64–67. <https://doi.org/10.1038/ng0597-64>.
- Limaye, Nisha, Laurence M. Boon, and Miikka Vikkula. 2009. "From Germline towards Somatic Mutations in the Pathophysiology of Vascular Anomalies." *Human Molecular Genetics* 18 (R1): R65–74. <https://doi.org/10.1093/hmg/ddp002>.
- Limaye, Nisha, Jaakko Kangas, Antonella Mendola, et al. 2015. "Somatic Activating PIK3CA Mutations Cause Venous Malformation." *American Journal of Human Genetics* 97 (6): 914–21. <https://doi.org/10.1016/j.ajhg.2015.11.011>.
- Lindhurst, Marjorie J., Julie C. Sapp, Jamie K. Teer, et al. 2011. "A Mosaic Activating Mutation in *AKT1* Associated with the Proteus Syndrome." *New England Journal of Medicine* 365 (7): 611–19. <https://doi.org/10.1056/NEJMoa1104017>.
- Luks, Valerie L., Nolan Kamitaki, Matthew P. Vivero, et al. 2015. "Lymphatic and Other Vascular Malformative/Overgrowth Disorders Are Caused by Somatic Mutations in PIK3CA." *The Journal of Pediatrics* 166 (4): 1048–54.e1-5. <https://doi.org/10.1016/j.jpeds.2014.12.069>.
- Luu, Maxime, Pierre Vabres, Aurélie Espitalier, et al. 2024a. "A Phase II Double-Blind Multicentre, Placebo-Controlled Trial to Assess the Efficacy and Safety of Alpelisib (BYL719) in Paediatric and Adult Patients with Megalencephaly-Capillary Malformation Polymicrogyria Syndrome (MCAP): The SESAM Study Protocol." *BMJ Open* 14 (12): e084614. <https://doi.org/10.1136/bmjopen-2024-084614>.
- Luu, Maxime, Pierre Vabres, Aurélie Espitalier, et al. 2024b. "A Phase II Double-Blind Multicentre, Placebo-Controlled Trial to Assess the Efficacy and Safety of Alpelisib (BYL719) in Paediatric and Adult Patients with Megalencephaly-Capillary Malformation Polymicrogyria Syndrome (MCAP): The SESAM Study Protocol." *BMJ Open* 14 (12): e084614. <https://doi.org/10.1136/bmjopen-2024-084614>.
- Macmurdo, Colleen F., Whitney Wooderchak-Donahue, Pinar Bayrak-Toydemir, et al. 2016. "RASA1 Somatic Mutation and Variable Expressivity in Capillary Malformation/Arteriovenous Malformation (CM/AVM) Syndrome." *American Journal of Medical Genetics Part A* 170 (6): 1450–54. <https://doi.org/10.1002/ajmg.a.37613>.
- Manevitz-Mendelson, Eugenia, Gil S. Leichner, Ortal Barel, et al. 2018. "Somatic NRAS Mutation in Patient with Generalized Lymphatic Anomaly." *Angiogenesis* 21 (2): 287–98. <https://doi.org/10.1007/s10456-018-9595-8>.
- Mansur, Ann, and Ivan Radovanovic. 2023. "Vascular Malformations: An Overview of Their Molecular Pathways, Detection of Mutational Profiles and Subsequent Targets for Drug Therapy." *Frontiers in Neurology* 14 (February). <https://doi.org/10.3389/fneur.2023.1099328>.
- Manzia, Tommaso Maria, Mario Carmellini, Paola Todeschini, et al. 2020. "A 3-Month, Multicenter, Randomized, Open-Label Study to Evaluate the Impact on Wound Healing of the Early (vs Delayed) Introduction of Everolimus in De Novo Kidney Transplant Recipients, With a Follow-up Evaluation at 12 Months After Transplant (NEVERWOUND Study)." *Transplantation* 104 (2): 374. <https://doi.org/10.1097/TP.0000000000002851>.
- Martins, Fabiana, Márcio Augusto de Oliveira, Qian Wang, et al. 2013. "A Review of Oral Toxicity Associated with mTOR Inhibitor Therapy in Cancer Patients." *Oral Oncology* 49 (4): 293–98. <https://doi.org/10.1016/j.oraloncology.2012.11.008>.
- Martins, Luciane, Priscila Alves Giovani, Pedro Diniz Rebouças, et al. 2017. "Computational Analysis for *GNAQ* Mutations: New Insights on the Molecular Etiology of Sturge-Weber Syndrome." *Journal of Molecular Graphics and Modelling* 76 (September): 429–40. <https://doi.org/10.1016/j.jmkgm.2017.07.011>.

- Maruani, Annabel, Elsa Tavernier, Olivia Boccara, et al. 2021. "Sirolimus (Rapamycin) for Slow-Flow Malformations in Children." *JAMA Dermatology* 157 (11): 1–10. <https://doi.org/10.1001/jamadermatol.2021.3459>.
- McDaniel, C. Griffin, Denise M. Adams, Kimberley E. Steele, et al. 2023. "Kaposiform Lymphangiomas: Diagnosis, Pathogenesis, and Treatment." *Pediatric Blood & Cancer* 70 (4): e30219. <https://doi.org/10.1002/pbc.30219>.
- Mincu, Raluca I., Amir A. Mahabadi, Lars Michel, et al. 2019. "Cardiovascular Adverse Events Associated With BRAF and MEK Inhibitors: A Systematic Review and Meta-Analysis." *JAMA Network Open* 2 (8): e198890. <https://doi.org/10.1001/jamanetworkopen.2019.8890>.
- Morin, Gabriel, Caroline Degrugillier-Chopin, Marie Vincent, et al. 2022. "Treatment of Two Infants with PIK3CA-Related Overgrowth Spectrum by Alpelisib." *Journal of Experimental Medicine* 219 (3): e20212148. <https://doi.org/10.1084/jem.20212148>.
- Morin, Gabriel, Alexandre P Garneau, Nabih Bouzakher, et al. 2025. "Somatic PIK3R1 Mutations in the iSH2 Domain Are Accessible to PI3K α Inhibition." *EMBO Molecular Medicine*, May 19, 1–19. <https://doi.org/10.1038/s44321-025-00249-9>.
- Nicholson, Cynthia L., Siobhan Flanagan, Michael Murati, et al. 2022. "Successful Management of an Arteriovenous Malformation with Trametinib in a Patient with Capillary-Malformation Arteriovenous Malformation Syndrome and Cardiac Compromise." *Pediatric Dermatology* 39 (2): 316–19. <https://doi.org/10.1111/pde.14912>.
- Nikolaev, Sergey I., Sandra Vetiska, Ximena Bonilla, et al. 2018. "Somatic Activating KRAS Mutations in Arteriovenous Malformations of the Brain." *New England Journal of Medicine* 378 (3): 250–61. <https://doi.org/10.1056/NEJMoa1709449>.
- Osborn, Alexander J., Peter Dickie, Derek E. Neilson, et al. 2015. "Activating PIK3CA Alleles and Lymphangiogenic Phenotype of Lymphatic Endothelial Cells Isolated from Lymphatic Malformations." *Human Molecular Genetics* 24 (4): 926–38. <https://doi.org/10.1093/hmg/ddu505>.
- Ours, Christopher A., Julie C. Sapp, Mia B. Hodges, Allison J. de Moya, and Leslie G. Biesecker. 2021. "Case Report: Five-Year Experience of AKT Inhibition with Miransertib (MK-7075) in an Individual with Proteus Syndrome." *Cold Spring Harbor Molecular Case Studies* 7 (6): a006134. <https://doi.org/10.1101/mcs.a006134>.
- Pagliuzzi, Angelica, Teresa Oranges, Giovanna Traficante, et al. 2021. "PIK3CA-Related Overgrowth Spectrum From Diagnosis to Targeted Therapy: A Case of CLOVES Syndrome Treated With Alpelisib." *Frontiers in Pediatrics* 9 (September): 732836. <https://doi.org/10.3389/fped.2021.732836>.
- Pascual, Julio, Stefan P. Berger, Oliver Witzke, et al. 2018. "Everolimus with Reduced Calcineurin Inhibitor Exposure in Renal Transplantation." *Journal of the American Society of Nephrology : JASN* 29 (7): 1979–91. <https://doi.org/10.1681/ASN.2018010009>.
- Penington, Anthony, Roderic J. Phillips, Nerida Sleebs, and Jane Halliday. 2023. "Estimate of the Prevalence of Vascular Malformations." *Journal of Vascular Anomalies* 4 (3): e068. <https://doi.org/10.1097/JOVA.000000000000068>.
- Petersen, T.A., L.A. Morrison, R.M. Schrader, and B.L. Hart. 2010. "Familial versus Sporadic Cavernous Malformations: Differences in Developmental Venous Anomaly Association and Lesion Phenotype." *AJNR: American Journal of Neuroradiology* 31 (2): 377–82. <https://doi.org/10.3174/ajnr.A1822>.
- Pithadia, D.J., A.M. Treichel, A.M. Jones, et al. 2020. "Dermatologic Adverse Events Associated with Use of Oral Mechanistic Target of Rapamycin Inhibitors in a Cohort of Individuals with Tuberous Sclerosis Complex." *The British Journal of Dermatology* 183 (3): 588–89. <https://doi.org/10.1111/bjd.19124>.
- Queisser, Angela, Emmanuel Seront, Laurence M. Boon, and Miikka Vikkula. 2021. "Genetic Basis and Therapies for Vascular Anomalies." *Circulation Research* 129 (1): 155–73. <https://doi.org/10.1161/CIRCRESAHA.121.318145>.

- Ramirez-Fort, Marigdalia K., Emily C. Case, Alyx C. Rosen, Felipe B. Cerci, Shenhong Wu, and Mario E. Lacouture. 2014. "Rash to the mTOR Inhibitor Everolimus: Systematic Review and Meta-Analysis." *American Journal of Clinical Oncology* 37 (3): 266. <https://doi.org/10.1097/COC.0b013e318277d62f>.
- Remy, Amandine, Albert Etingin, Paul Gavra, et al. 2025. "Fixed Dosing of Alpelisib for Children with Vascular Anomalies: Can We Do Better?" *British Journal of Clinical Pharmacology* 91 (3): 914–20. <https://doi.org/10.1111/bcp.16388>.
- Remy, Amandine, Thai Hoa Tran, Josée Dubois, et al. 2022. "Repurposing Alpelisib, an Anti-Cancer Drug, for the Treatment of Severe TIE2-Mutated Venous Malformations: Preliminary Pharmacokinetics and Pharmacodynamic Data." *Pediatric Blood & Cancer* 69 (10): e29897. <https://doi.org/10.1002/pbc.29897>.
- Ren, Aileen A., Daniel A. Snellings, Sophie Y. Su, et al. 2021. "PIK3CA and CCM Mutations Fuel Cavernomas through a Cancer-like Mechanism." *Nature* 594 (7862): 271–76. <https://doi.org/10.1038/s41586-021-03562-8>.
- Research, Center for Drug Evaluation and. 2021. "FDA Grants Accelerated Approval to Sotorasib for KRAS G12C Mutated NSCLC." *FDA*, June 11. <https://www.fda.gov/drugs/resources-information-approved-drugs/fda-grants-accelerated-approval-sotorasib-kras-g12c-mutated-nsclc>.
- Research, Center for Drug Evaluation and. 2022. "FDA Grants Accelerated Approval to Adagrasib for KRAS G12C-Mutated NSCLC." *FDA*, December 12. <https://www.fda.gov/drugs/resources-information-approved-drugs/fda-grants-accelerated-approval-adagrasib-kras-g12c-mutated-nsclc>.
- Revencu, Nicole, Laurence M. Boon, Antonella Mendola, et al. 2013. "RASA1 Mutations and Associated Phenotypes in 68 Families with Capillary Malformation–Arteriovenous Malformation." *Human Mutation* 34 (12): 1632–41. <https://doi.org/10.1002/humu.22431>.
- Ryu, Jeong Yeop, Yong June Chang, Joon Seok Lee, et al. 2023. "A Nationwide Cohort Study on Incidence and Mortality Associated with Extracranial Vascular Malformations." *Scientific Reports* 13 (1): 13950. <https://doi.org/10.1038/s41598-023-41278-z>.
- Sabata, Helena, Ariadna Roca-Coll, Jose A. Dengra, et al. 2025. "Context-Dependent Response of Endothelial Cells to PIK3CA Mutation." Preprint, bioRxiv, February 25. <https://doi.org/10.1101/2025.02.25.640041>.
- Salem, Mohamed E., Sherif M. El-Refai, Wei Sha, et al. 2022. "Landscape of KRASG12C, Associated Genomic Alterations, and Interrelation With Immuno-Oncology Biomarkers in KRAS-Mutated Cancers." *JCO Precision Oncology*, no. 6 (March): e2100245. <https://doi.org/10.1200/PO.21.00245>.
- Salo-Mullen, Erin E., Jinru Shia, Isaac Brownell, et al. 2014. "Mosaic Partial Deletion of the PTEN Gene in a Patient with Cowden Syndrome." *Familial Cancer* 13 (3): 459–67. <https://doi.org/10.1007/s10689-014-9709-4>.
- Schmidt, M., K. Lübbe, T. Decker, et al. 2022. "A Multicentre, Randomised, Double-Blind, Phase II Study to Evaluate the Tolerability of an Induction Dose Escalation of Everolimus in Patients with Metastatic Breast Cancer (DESIREE)." *ESMO Open* 7 (6): 100601. <https://doi.org/10.1016/j.esmoop.2022.100601>.
- Schönewolf-Greulich, Bitten, Helena Gásdal Karstensen, Tina D. Hjortshøj, et al. 2022. "Early Diagnosis Enabling Precision Medicine Treatment in a Young Boy with PIK3R1-Related Overgrowth." *European Journal of Medical Genetics* 65 (10): 104590. <https://doi.org/10.1016/j.ejmg.2022.104590>.
- Seebauer, Caroline T., Benedikt Wiens, Constantin A. Hintschich, et al. 2024. "Targeting the Microenvironment in the Treatment of Arteriovenous Malformations." *Angiogenesis* 27 (1): 91–103. <https://doi.org/10.1007/s10456-023-09896-3>.
- Seront, Emmanuel, Laurence M. Boon, and Miikka Vikkula. 1993. "TEK-Related Venous Malformations." In *GeneReviews®*, edited by Margaret P. Adam, Jerry Feldman, Ghayda M. Mirzaa, Roberta A. Pagon, Stephanie E. Wallace, and Anne Amemiya. University of Washington, Seattle. <http://www.ncbi.nlm.nih.gov/books/NBK1967/>.
- Seront, Emmanuel, An Van Damme, Catherine Legrand, et al. 2023. "Preliminary Results of the European Multicentric Phase III Trial Regarding Sirolimus in Slow-Flow Vascular Malformations." *JCI Insight* 8 (21). <https://doi.org/10.1172/jci.insight.173095>.

- Servattalab, Sarah, and Pierre-Olivier Grenier. 2025. "Tufted Angioma with Kasabach–Merritt Phenomenon." *New England Journal of Medicine* 392 (20): e51. <https://doi.org/10.1056/NEJMicm2414170>.
- Skoulidis, Ferdinandos, Bob T. Li, Grace K. Dy, et al. 2021. "Sotorasib for Lung Cancers with KRAS p.G12C Mutation." *New England Journal of Medicine* 384 (25): 2371–81. <https://doi.org/10.1056/NEJMoa2103695>.
- Snellings, Daniel A., Romuald Girard, Rhonda Lightle, et al. 2022. "Developmental Venous Anomalies Are a Genetic Primer for Cerebral Cavernous Malformations." *Nature Cardiovascular Research* 1 (March): 246–52. <https://doi.org/10.1038/s44161-022-00035-7>.
- Snellings, Daniel A., Courtney C. Hong, Aileen A. Ren, et al. 2021. "Cerebral Cavernous Malformation: From Mechanism to Therapy." *Circulation Research* 129 (1): 195–215. <https://doi.org/10.1161/CIRCRESAHA.121.318174>.
- Sterba, Martin, Petra Pokorna, Renata Faberova, et al. 2023a. "Targeted Treatment of Severe Vascular Malformations Harboring PIK3CA and TEK Mutations with Alpelisib Is Highly Effective with Limited Toxicity." *Scientific Reports* 13 (1): 10499. <https://doi.org/10.1038/s41598-023-37468-4>.
- Sterba, Martin, Petra Pokorna, Renata Faberova, et al. 2023b. "Targeted Treatment of Severe Vascular Malformations Harboring PIK3CA and TEK Mutations with Alpelisib Is Highly Effective with Limited Toxicity." *Scientific Reports* 13 (1): 10499. <https://doi.org/10.1038/s41598-023-37468-4>.
- Strickler, John H., Hironaga Satake, Thomas J. George, et al. 2023. "Sotorasib in KRAS p.G12C–Mutated Advanced Pancreatic Cancer." *New England Journal of Medicine* 388 (1): 33–43. <https://doi.org/10.1056/NEJMoa2208470>.
- Tedesco-Silva, Helio, Faouzi Saliba, Markus J. Barten, et al. 2022. "An Overview of the Efficacy and Safety of Everolimus in Adult Solid Organ Transplant Recipients." *Transplantation Reviews* 36 (1): 100655. <https://doi.org/10.1016/j.trre.2021.100655>.
- Thomas, Markus, and Hellmut G. Augustin. 2009. "The Role of the Angiopoietins in Vascular Morphogenesis." *Angiogenesis* 12 (2): 125–37. <https://doi.org/10.1007/s10456-009-9147-3>.
- Venot, Quitterie, Thomas Blanc, Smail Hadj Rabia, et al. 2018a. "Targeted Therapy in Patients with PIK3CA-Related Overgrowth Syndrome." *Nature* 558 (7711): 540–46. <https://doi.org/10.1038/s41586-018-0217-9>.
- Venot, Quitterie, Thomas Blanc, Smail Hadj Rabia, et al. 2018b. "Targeted Therapy in Patients with PIK3CA-Related Overgrowth Syndrome." *Nature* 558 (7711): 540–46. <https://doi.org/10.1038/s41586-018-0217-9>.
- Villanueva, Bernat, Adriana Iriarte, Raquel Torres-Iglesias, Miriam Muñoz Bolaño, Pau Cerdà, and Antoni Riera-Mestre. 2023. "Aflibercept for Gastrointestinal Bleeding in Hereditary Hemorrhagic Telangiectasia: A Case Report." *Medicina* 59 (9): 9. <https://doi.org/10.3390/medicina59091533>.
- Wang, Diana G., Dulce M. Barrios, Victoria S. Blinder, et al. 2020. "Dermatologic Adverse Events Related to the PI3K α Inhibitor Alpelisib (BYL719) in Patients with Breast Cancer." *Breast Cancer Research and Treatment* 183 (1): 227–37. <https://doi.org/10.1007/s10549-020-05726-y>.
- Wang, Dun, Lingling Ge, Zizhen Guo, et al. 2022. "Efficacy and Safety of Trametinib in Neurofibromatosis Type 1-Associated Plexiform Neurofibroma and Low-Grade Glioma: A Systematic Review and Meta-Analysis." *Pharmaceuticals* 15 (8): 956. <https://doi.org/10.3390/ph15080956>.
- Wang, Yang-Yang, Li-Ping Zou, Kai-Feng Xu, et al. 2024. "Long-Term Safety and Influence on Growth in Patients Receiving Sirolimus: A Pooled Analysis." *Orphanet Journal of Rare Diseases* 19 (1): 299. <https://doi.org/10.1186/s13023-024-03243-5>.
- Yabu, Takeshi, Hidekazu Tomimoto, Yoshimitsu Taguchi, Shohei Yamaoka, Yasuyuki Igarashi, and Toshiro Okazaki. 2005. "Thalidomide-Induced Antiangiogenic Action Is Mediated by Ceramide through Depletion of VEGF Receptors, and Is Antagonized by Sphingosine-1-Phosphate." *Blood* 106 (1): 125–34. <https://doi.org/10.1182/blood-2004-09-3679>.

Yuan, Hai Tao, Eliyahu V. Khankin, S. Ananth Karumanchi, and Samir M. Parikh. 2009. "Angiopoietin 2 Is a Partial Agonist/Antagonist of Tie2 Signaling in the Endothelium." *Molecular and Cellular Biology* 29 (8): 2011–22. <https://doi.org/10.1128/MCB.01472-08>.

Zabeida, Alexandra, Jack J. Brzezinski, Jonathan D. Wasserman, et al. 2024. "Sirolimus for Vascular Anomalies Associated with PTEN Hamartoma Tumor Syndrome." *Pediatric Blood & Cancer* 71 (11): e31282. <https://doi.org/10.1002/pbc.31282>.

Zerbib, Lola, Sophia Ladraa, Antoine Fraissenon, et al. 2024. "Targeted Therapy for Capillary-Venous Malformations." *Signal Transduction and Targeted Therapy* 9 (1): 1–16. <https://doi.org/10.1038/s41392-024-01862-9>.

Zhou, Zinan, Alan T. Tang, Weng-Yew Wong, et al. 2016. "Cerebral Cavernous Malformations Arise from Endothelial Gain of MEKK3-KLF2/4 Signalling." *Nature* 532 (7597): 122–26. <https://doi.org/10.1038/nature17178>.

23rd Oct 2025

Dear Prof. Canaud, Dear Guillaume,

Thank you for submitting your revised manuscript to EMBO Molecular Medicine. Please accept my apologies for the delay in responding; one referee required additional time to complete their report. We have now received feedback from referees #2 and #3 on your revised manuscript. As you will see below, they appreciated the work done on the revisions but nevertheless have a few remaining suggestions.

We would therefore like to invite you to make further revisions to address these minor concerns.

Please also address the following editorial issues:

- Please indicate in track changes mode any new modification to the text.
- Please provide up to 5 keywords.
- Funding should be merged with Acknowledgements. Please add the entire list of funders, with project numbers where available to our system. This list will be linked upon publication, so it is essential that it accurately reflects all funding information.
- Please change "Competing interests" to "Disclosure and competing interests statement" and add "G.C. is an editorial advisory board member."
- The references format should be corrected to 10 author names listed before et al; please remove the DOIs.
- Please include a Glossary: EMBO Molecular Medicine articles are accompanied by a glossary explaining some of the terms used for laymen. Glossary is not a list of abbreviations; abbreviations should be defined the first time they are used in the text.
- Please include a section termed Pending issues: At the end of each article, there is a box highlighting issues that still need further studies and where research efforts should converge. This should be limited to a few points.
- Figures:
 - o If there are certain aspects of your figures that are based upon assumptions or where the scientific data remains ambiguous, please add a comment so that we can work with you on an accurate depiction. Please ensure the directionality and nature of interactions is presented accurately.
 - o If the figure or single panels of the figure have been adapted from a published figure, please add this information to the figure legend (e.g., 'Adapted from...' or 'Based on...').
 - o Please only re-use figures or parts of a figure if this is essential for understanding the concept communicated. If the figure contains re-used images or elements of images, please make sure that you have the permission/license to publish it (this also applies to your own previous work, if the journal you published in retains copyright. Certain 'creative commons' open access licenses, such as CC-BY 4.0, allow re-use without additional formal permissions). All re-used material must be explicitly cited.
 - o If you use an image data base for scientific iconography (e.g., BioRender), please let us know if you have a license that allows for publication in an academic journal. Often authors use misleading iconography for expedience. Please ensure the information shown is scientifically accurate.

Looking forward to receiving your revised manuscript,

With kind regards,

Lise

Lise Roth, Ph.D.
Senior Editor
EMBO Molecular Medicine

***** Reviewer's comments *****

Referee #2 (Remarks for Author):

In this revised version, Morin et al. have significantly enhanced the quality of their review article, incorporating comprehensive feedback from reviewers.

Thank you for your insightful comments on alpelisib. In our pediatric patient population, growth delay is increasingly concerning, as dose escalation (even beyond 3 mg/kg/d) is often necessary for clinical efficacy. This is evidenced by this newly published

paper that should be cited: Triana P et al, JOVA, 2025 (DOI: 10.1097/JOVA.000000000000115). Unpublished real-world data on alpelisib-induced growth delay from other institutions were presented at two major vascular anomalies conferences this year and are in preparation for publication. It appears that growth delay with alpelisib is underreported rather than rare. I encourage the authors to address this in the alpelisib section to contribute to an unbiased and objective report of global observations. Similarly, our experience shows that the impact of alpelisib on glucose metabolism is not rare and occurs independently of confounding, common risk factors for DM2 or regional variations. Including a note about the association of alpelisib treatment with growth retardation and DM2 in preliminary findings would enhance the review's credibility in reporting real-world data.

Regarding KMP in KLA, our hematological understanding confirms the occurrence of KMP in KLA. Initially, thrombocytopenia and decreased fibrinogen levels may be milder compared to KHE and TA, leading us to term it a KMP-like phenotype. However, this can progress to a full clinical DIC picture with life-threatening hemorrhage. We have lost genetically (activating somatic NRAS variant) and histologically confirmed KLA patients to a typical KMP > DIC > massive fatal hemorrhage sequence earlier. I suggest including the KMP-like phenotype in KLA, which may initially appear milder but can become severe, as outlined. Please add citations such as PMID: 35385405, which in the abstracts notes an aggressive clinical course marked by hemorrhage, thrombocytopenia, diminished fibrinogen levels, and a 21% mortality rate. For Figure 1, please specify VEGFR (likely VEGFR2) and ANG (likely ANG1). The omission of the nucleus is a common oversight in the field, neglecting transcriptional biological accuracy. The figure design is the author's prerogative, and it is commendable to see growth, proliferation, and differentiation depicted, as it enhances biological precision. Additionally, a legend for the symbols used would be beneficial (I could not find it) .

Figure 2 and the accompanying table show significant improvement in representative AVM imaging and the table's quality. As a comment rather than suggestion for inclusion into the manuscript: PIK3CA has occurred in AVMs at our institution with VAF >5% (including in re-testing) which was surprising to us considering the fast-flow entity. Although rare, it illustrates the broad spectrum of genetic variants in AVM beyond RAS signaling.

I am pleased to see that the citation issues have been addressed, a critical quality aspect especially of a review article. Thank you for the additional comments, which I found valuable.

Referee #3 (Remarks for Author):

The review has greatly improved.

Yet, some references and key points should be included in the review:

Authors should include the manuscript recently published by the Makinen group on the feedback loop of PI3K/ANG2 in venous malformations (PMID: 40410415)

I still think authors should cite Manevitz-Mendelson et al. (Angiogenesis 2018) when discussing NRAS Q61R

When discussing AKT inhibitors, authors should discuss their use in KLA, as recently shown (PMID: 40838953)

mTORC2 phosphorylate AKT but it is not activated by PIP3. Please rephrase in line 150.

In line 187, authors should cite 2 important papers in the context of LMs, PMID: 32513927 and PMID: 36688917

Editors' Recommendations

Please also address the following editorial issues:

- Please indicate in track changes mode any new modification to the text.

Done

- Please provide up to 5 keywords.

Done

- Funding should be merged with Acknowledgements. Please add the entire list of funders, with project numbers where available to our system. This list will be linked upon publication, so it is essential that it accurately reflects all funding information.

Done

- Please change "Competing interests" to "Disclosure and competing interests statement" and add "G.C. is an editorial advisory board member."

Done

- The references format should be corrected to 10 author names listed before et al; please remove the DOIs.

Done

- **Please include a Glossary:** EMBO Molecular Medicine articles are accompanied by a glossary explaining some of the terms used for laymen. Glossary is not a list of abbreviations; abbreviations should be defined the first time they are used in the text.

Done

- Please include a **section termed Pending issues:** At the end of each article, there is a box highlighting issues that still need further studies and where research efforts should converge. This should be limited to a few points.

Done

- **Figures:**

o If there are certain aspects of your figures that are based upon assumptions or where the scientific data remains ambiguous, please add a comment so that we can work with you on an accurate depiction. Please ensure the directionality and nature of interactions is presented accurately.

OK

o If the figure or single panels of the figure have been adapted from a published figure, please add this information to the figure legend (e.g., 'Adapted from...' or 'Based on...').

OK

o Please only re-use figures or parts of a figure if this is essential for understanding the concept communicated. If the figure contains re-used images or elements of images, please make sure that you have the permission/license to publish it (this also applies to your own previous work, if the journal you published in retains copyright. Certain 'creative commons' open access licenses, such as CC-BY 4.0, allow re-use without additional formal permissions). All re-used material must be explicitly cited.

OK

o If you use an image data base for scientific iconography (e.g., BioRender), please let us know if you have a license that allows for publication in an academic journal. Often authors use misleading iconography for expedience. Please ensure the information shown is scientifically accurate.

License obtained and submitted to the journal.

Looking forward to receiving your revised manuscript,

With kind regards,

Lise

Lise Roth, Ph.D.

Senior Editor

EMBO Molecular Medicine

Referees' comments

Referee #2 (Remarks for Author):

In this revised version, Morin et al. have significantly enhanced the quality of their review article, incorporating comprehensive feedback from reviewers.

Thank you for your insightful comments on alpelisib. In our pediatric patient population, growth delay is increasingly concerning, as dose escalation (even beyond 3 mg/kg/d) is often necessary for clinical efficacy. This is evidenced by this newly published paper that should be cited: Triana P et al, JOVA, 2025 (DOI: 10.1097/JOVA.000000000000115).

Unpublished real-world data on alpelisib-induced growth delay from other institutions were presented at two major vascular anomalies conferences this year and are in preparation for publication. It appears that growth delay with alpelisib is underreported rather than rare. I encourage the authors to address this in the alpelisib section to contribute to an unbiased and objective report of global observations. Similarly, our experience shows that the impact of alpelisib on glucose metabolism is not rare and occurs independently of confounding, common risk factors for DM2 or regional variations. Including a note about the association of alpelisib treatment with growth retardation and DM2 in preliminary findings would enhance the review's credibility in reporting real-world data.

As requested, we have added the study by Triana et al. to the references and discussed it in more detail. However, while we do not contest the reviewers' point of view, this has not been our experience, nor is it supported by the findings from the EPIK P1, EPIK P2, EPIK P3, and EPIK L1 clinical trials. In all these studies, an independent safety board reviewed adverse events, with particular attention to the effects of alpelisib on growth, puberty, and fertility.

We are also about to report the long-term outcomes of more than 50 pediatric patients treated with alpelisib. Growth and hormonal parameters have been closely monitored in collaboration with endocrinologists. The key message is that we have not observed any significant impact of the drug on growth or hormonal function.

Regarding KMP in KLA, our hematological understanding confirms the occurrence of KMP in KLA. Initially, thrombocytopenia and decreased fibrinogen levels may be milder compared to KHE and TA, leading us to term it a KMP-like phenotype. However, this can progress to a full clinical DIC picture with life-threatening hemorrhage. We have lost genetically (activating somatic NRAS variant) and histologically confirmed KLA patients to a typical KMP > DIC >

massive fatal hemorrhage sequence earlier. I suggest including the KMP-like phenotype in KLA, which may initially appear milder but can become severe, as outlined. Please add citations such as PMID: 35385405, which in the abstracts notes an aggressive clinical course marked by hemorrhage, thrombocytopenia, diminished fibrinogen levels, and a 21% mortality rate.

We thank the Reviewer for the insightful discussion. We have also observed severe DIC in a young patient with NRAS^{Q61R}-related KLA. We agree with the Reviewer that the term “KMP-like phenotype” may better suit this specific coagulation disorder – as its pathophysiology may differ from bona fide KMP associated with vascular tumors. We modified the main text as follows:

KLA exhibits the poorest prognosis among CLAs, with high mortality rates (Eng et al, 2018; Perez-Atayde et al, 2022). Similar to the Kasabach-Merritt phenomenon (KMP) – a severe localized thrombotic process leading to consumptive coagulopathy in vascular tumors such as kaposiform hemangioendothelioma – KLA often presents with a KMP-like syndrome resulting in life-threatening DIC (Kelly, 2010; Perez-Atayde et al, 2022; Servattalab & Grenier, 2025). KLA is almost exclusively driven by the Q61R variant of NRAS, suggesting that lymphatic homeostasis relies heavily on this isoform (Manevitz-Mendelson et al, 2018; Barclay et al, 2019; Ozeki et al, 2019; Li et al, 2023a; Ramwani et al, 2023).

For Figure 1, please specify VEGFR (likely VEGFR2) and ANG (likely ANG1). The omission of the nucleus is a common oversight in the field, neglecting transcriptional biological accuracy. The figure design is the author's prerogative, and it is commendable to see growth, proliferation, and differentiation depicted, as it enhances biological precision. Additionally, a legend for the symbols used would be beneficial (I could not find it).

As requested, we corrected “VEGFR” to “VEGFR2” and “ANG” to “ANG1”.

The following sentence was already included in Figure 1 legend: “Crosses and lightning bolts denote components with loss-of-function and gain-of-function variants reported in vascular malformations, respectively.”

We have added the following indication for clarity: “Sharp arrows indicate direct (solid line) or indirect (dashed line) activation. Blunt arrows show inhibition.”

Figure 2 and the accompanying table show significant improvement in representative AVM imaging and the table's quality. As a comment rather than suggestion for inclusion into the manuscript: PIK3CA has occurred in AVMs at our institution with VAF >5% (including in re-testing) which was surprising to us considering the fast-flow entity. Although rare, it illustrates the broad spectrum of genetic variants in AVM beyond RAS signaling.

We thank the Reviewer for their valuable comment. We observed a similar association in a patient with brain AVM and a post-zygotic *PIK3CA* variant in a skin biopsy, though the association may be fortuitous, rendering the causality of *PIK3CA* uncertain. We did not include this specificity in the main text for clarity, and because the evidence for *PIK3CA* involvement in AVMs is scarce – though this rare association is definitely of interest and questions the current PI3K/MAPK pathway dichotomy.

I am pleased to see that the citation issues have been addressed, a critical quality aspect especially of a review article.

Thank you for the additional comments, which I found valuable.

We sincerely thank the reviewer for their positive feedback. We are glad that peer-review significantly improved our manuscript.

Referee #3 (Remarks for Author):

The review has greatly improved.

We thank the reviewer for their positive feedback.

Yet, some references and key points should be included in the review:

Authors should include the manuscript recently published by the Makinen group on the feedback loop of PI3K/ANG2 in venous malformations (PMID: 40410415)

As suggested, we have added the Makinen group's references to the manuscript:

"Additionally, recent work proposed that a PI3K-ANG2-TIE2 amplification loop may participate in the growth of venous malformations. In PIK3CA-mutant ECs, AKT-mediated FOXO1 inhibition lowers ANG2 synthesis. The resulting ANG1/ANG2 imbalance may account for the increased TIE2 phosphorylation seen in these lesions. This model is further supported by the finding that TIE2 inhibition – despite acting upstream of PIK3CA – reduces VM growth in vivo (Kraft et al, 2025)."

I still think authors should cite Manevitz-Mendelson et al. (Angiogenesis 2018) when discussing NRAS Q61R

As requested, we have added the reference to the manuscript.

When discussing **AKT inhibitors**, authors should discuss their use in **KLA**, as recently shown (PMID: **40838953**)

As requested, we have added the cited reference to the manuscript:

"Finally, AKT inhibitors have recently showed preclinical efficacy in a zebrafish model of KLA, suggesting that disorders involving both the PI3K and MAPK pathways might be accessible to similar therapies (Bassi et al, 2025)."

mTORC2 phosphorylate AKT but it is not activated by PIP3. Please rephrase in line 150.

We thank the reviewer for identifying this mistake. We have changed the text accordingly:

"PIP3 activates Phosphoinositide-Dependent Kinase 1 (PDK1), which, along with mammalian Target Of Rapamycin 2 (mTORC2), activates the serine/threonine kinase AKT."

In line 187, authors should cite **2 important papers in the context of LMs, PMID: 32513927 and PMID: 36688917**

As proposed, we have added the cited references to the manuscript, which echo the mechanisms discussed in the venous malformation section:

"Similar to the amplification loop described in venous malformations, VEGF-C promotes the growth of PIK3CA-mutated LMs by further activating the PI3K pathway (Martinez-Corral et al, 2020).

Additionally, endothelial-macrophage crosstalk may reinforce this process: PIK3CA-mutated lymphatic ECs were shown to secrete chemokines that recruit macrophages, which in turn produce VEGF-C and sustain lymphangiogenesis (Petkova et al, 2023).”

10th Nov 2025

Dear Prof. Canaud, Dear Guillaume,

Thank you for submitting your revised files. I am pleased to inform you that your manuscript is accepted for publication and is now being sent to our publisher to be included in the next available issue of EMBO Molecular Medicine!

Your manuscript will be processed for publication by EMBO Press. It will be copy edited and you will receive page proofs prior to publication. Please note that you will be contacted by Springer Nature Author Services to complete licensing information.

This Review is free of charge. When you are contacted in a few weeks to sign your license agreement and review article proofs, please enter the following token into the relevant field in the Springer Nature author services system: [removed].

If you have any questions, please do not hesitate to contact the Editorial Office.
Thank you for your nice contribution to EMBO Molecular Medicine!

With kind regards,

Lise
